# Miniaturized spectrometer with intrinsic long-term image memory

Gang Wu[1], Mohamed Abid[1], Mohamed Zerara [2], Jiung Cho[3,4], Miri Choi [5], Cormac Ó Coileáin[6], Kuan-Ming Hung [7], Ching-Ray Chang [8,9], Igor V. Shvets[10] & Han-Chun Wu [1]✉

Miniaturized spectrometers have great potential for use in portable optoelectronics and wearable sensors. However, current strategies for miniaturization rely on von Neumann architectures, which separate the spectral sensing, storage, and processing modules spatially, resulting in high energy consumption and limited processing speeds due to the storage-wall problem. Here, we present a miniaturized spectrometer that utilizes a single $SnS_2/ReSe_2$ van der Waals heterostructure, providing photodetection, spectrum reconstruction, spectral imaging, long-term image memory, and signal processing capabilities. Interface trap states are found to induce a gate-tunable and wavelength-dependent photogating effect and a non-volatile optoelectronic memory effect. Our approach achieves a footprint of 19 μm, a bandwidth from 400 to 800 nm, a spectral resolution of 5 nm, and a $> 10^4$ s long-term image memory. Our single-detector computational spectrometer represents a path beyond von Neumann architectures.

Spectrometers play a crucial role in scientific research and numerous industrial applications[1,2]. However, traditional spectrometers typically comprise bulky mechanical parts such as optical gratings and Michelson interferometers[3]. Therefore, the miniaturization of spectrometers while retaining high spectral resolution, broad spectrum sensing, and fast response, at low cost, is a subject of intense research interest due to potential applications in portable and wearable optoelectronics[4–15]. Common spectrometer miniaturization strategies rely on integrated detector arrays and separate optical elements with wavelength-dependent optical properties, including compact interferometers[4,12,16], quantum dots[5,12], photonic crystals[8,10], and metasurfaces[6,13,17]. Despite recent advances, miniaturizing these spectrometers down to the micrometer scale remains challenging due to limitations imposed by the optical path length[11]. Recently, a computational spectroscopy technology that leverages reconstructive

mathematical algorithms has been established, allowing for the fabrication of detector-only spectrometers without separate optics. These approaches include various photodetectors and photodetector arrays such as bandgap engineered nanowire arrays[14,18], and two-dimensional (2D) van der Waals (vdW) materials[7] and heterostructures[11,15] with gate-tunable and wavelength-dependent photoresponsivity. However, the commercialization and practical applications of miniaturized spectrometers and imaging systems rely on their integration with silicon chips[19,20], which are often constructed with von Neumann architectures[21] where the spectral sensing, storage, and processing modules are spatially separated[22]. This leads to high energy consumption[23] and slow processing speeds due to the storage-wall problem[24].

In this work, we demonstrate the possibility of fabricating a miniaturized spectrometer using a single $SnS_2/ReSe_2$ vdW

[1]School of Physics, Beijing Institute of Technology, Beijing 100081, P. R. China. [2]University of Applied Sciences, Geneva, Switzerland. [3]Western Seoul Cente, Korea Basic Science Institute, Seoul 03579, Republic of Korea. [4]Department of Advanced Materials Engineering, Chung-Ang University, 4726, Seodong-daero, Daedeok-myeon, Anseong-si, Gyeonggi-do 17546, Republic of Korea. [5]Chuncheon Center, Korea Basic Science Institute, Chuncheon 24341, Republic of Korea. [6]Institute of Physics, Faculty of Electrical Engineering and Information Technology, University of the Bundeswehr Munich, Neubiberg 85577, Germany. [7]Department of Electronics Engineering, National Kaohsiung University of Science and Technology, Kaohsiung 807, Taiwan, ROC. [8]Quantum Information Center, Chung Yuan Christian University, Taoyuan 32023, Taiwan, ROC. [9]Department of Physics, National Taiwan University, Taipei 106, Taiwan, ROC. [10]School of Physics, Trinity College Dublin, Dublin, Dublin 2, Ireland. ✉e-mail: wuhc@bit.edu.cn

heterostructure with the capabilities of photodetection, spectrum reconstruction, spectral imaging, and detectivity within the visible spectral range. By leveraging the gate-tunable photoresponse, reconstructive algorithms, and compressed sensing methodologies, we demonstrate a miniaturized spectrometer and spectral imager with a footprint of merely 19 μm, a bandwidth from 400 to 800 nm, a spectral resolution of 5 nm, and a $10^4$ s long-term image memory. Additionally, our miniaturized spectrometer not only serves as storage but also has processing capabilities when alternate laser and gate pulses are applied. Our approach represents a significant step towards single-detector computational spectrometers beyond von Neumann architectures.

## Results

### Device arrangement and photocurrent mechanism

Figure 1a displays a vertically stacked $SnS_2$/$ReSe_2$ heterostructure, used in this experiment with an active region of approximately 28-nm-thick $SnS_2$ and 40-nm-thick $ReSe_2$ (Supplementary Fig. 1). The vdW flakes were mechanically exfoliated from high quality single crystals, verified by Raman spectroscopy and high-resolution transmission electron microscopy (HRTEM) characterization (Fig. 1b–d). The overlapped region (green plot) exhibited both materials' characteristic Raman peaks, indicating the presence of two distinct materials[25–28]. The contact interface showed a typical type-II band structure (Supplementary Figs. 2 and 3) and the individual $SnS_2$ and $ReSe_2$ both showed

strong light absorption within the visible spectrum range (Supplementary Fig. 4), resulting a strong photoresponse to visible light. Our vdW heterostructure device achieves a large photocurrent ($I_{ph}$) of approximately 0.6 μA (Fig. 1e), with the highest photoresponsivity ($R$) calculated to be approximately 200 A/W at a gate voltage ($V_g$) of 50 V and an incident laser power density ($P_{in}$) of 0.22 mW/cm² (Supplementary Fig. 5). The device exhibited a high specific detectivity (D*) of $3.4 \times 10^{12}$ cm $Hz^{1/2}$ $W^{-1}$ due to its ultralow dark current in the off state ($V_g < 2$ V, Supplementary Fig. 5). The photocurrent of our device increased linearly with incident light power in the low power density range (<0.8 mW/cm²), and its slope depends on the gate voltage (Fig. 1g), laying an important foundation for its application in spectrometry, as discussed later[14].

Interestingly, photocurrent peaks were observed in the spectrum, which moved from ~580 nm to ~670 nm when the gate voltage was increased from −20 V to 20 V (Fig. 1f). Scanning photocurrent maps under 532, 633, and 785 nm laser illuminations for a fixed laser power of 0.5 mW (Fig. 2a) showed that the $SnS_2$ and $ReSe_2$ regions' photocurrent decreased with λ, while the photocurrent in the junction area exhibited a peak (Fig. 2b–d). This indicates that the $SnS_2$ and $ReSe_2$ interface dominates the photon absorption and energy conversion process. Moreover, the photocurrent of the junction area is even greater than the sum of the photocurrent in the $SnS_2$ and $ReSe_2$ regions, pointing to the junction's essential role in establishing the photocurrent. TEM analysis and XPS characterization suggest that our $SnS_2$ possesses ~2% sulfur

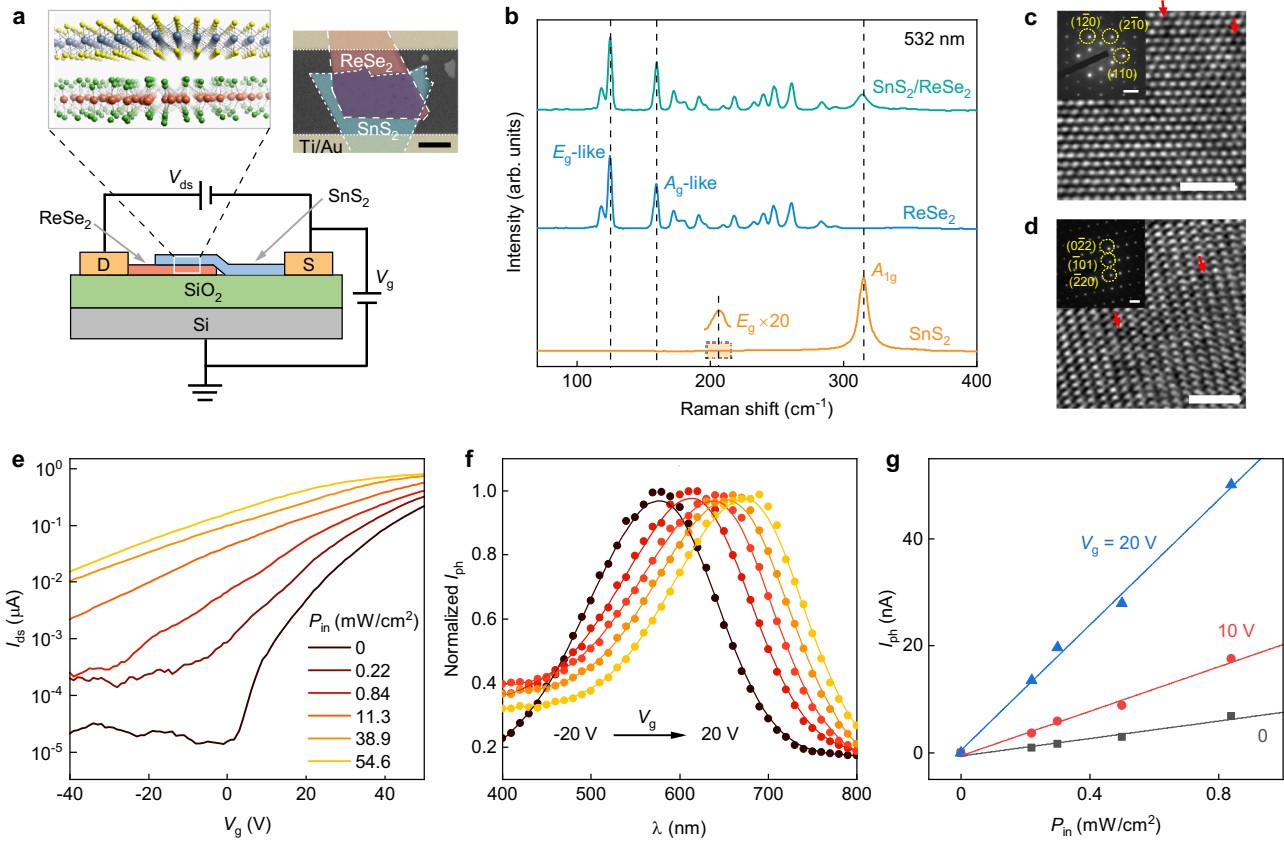

**Fig. 1 | Characterization of van der Waals (vdW) spectrometers. a** Schematic of the vertically stacked $SnS_2$/$ReSe_2$ vdW heterostructure device on a gated Si/$SiO_2$ substrate. $V_{ds}$: drain-source voltage; $V_g$: gate voltage. In the upper right is an optical image of the device with a 5 μm scale bar. **b** Raman spectra of a $SnS_2$/$ReSe_2$ heterostructure in different regions. The vertical dashed lines highlight the positions of corresponding Raman peaks. **c** High-resolution transmission electron microscopy (HRTEM) image of mechanically exfoliated $SnS_2$ and (**d**) $ReSe_2$ layers, where red arrows highlight the positions of vacancies. The insets show the corresponding

selected area electron diffraction (SAED) images. Scale bar for HRTEM, 2 nm. Scale bar for SAED, 2 nm⁻¹. **e** Transfer curves of the device for 635 nm laser illumination with different incident power densities ($P_{in}$) at $V_{ds} = 1$ V. **f** Photocurrent ($I_{ph}$) as a function of light wavelength at various gate voltages from −20 V to 20 V. The $I_{ph}$ are normalized to the maximum values at each $V_g$. Solid lines are guides to the eye (adjacent-averaging smoothing of the data points). **g** Photocurrent of the device as a function of laser power density at different gate voltages. Solid lines are linear fittings to the corresponding data points.

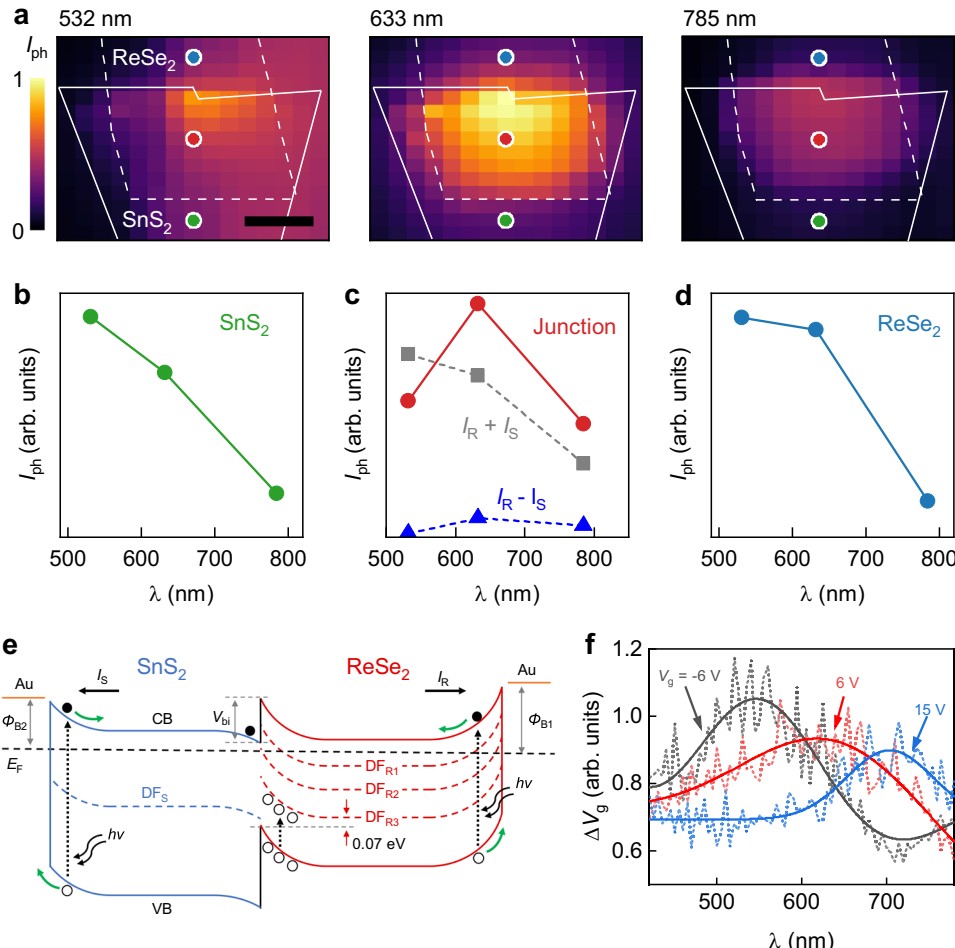

**Fig. 2 | Mechanism of photocurrent. a** Scanning photocurrent maps of the heterostructure under 532, 635, and 785 nm (from left to right) laser illumination. Solid and dashed lines show the outlines of $SnS_2$ and $ReSe_2$, respectively. Colored circles indicate typical $ReSe_2$ (blue), heterojunction (red), and $SnS_2$ (green) regions, respectively. Scale bar, 5 μm. **b**–**d** Photocurrent in the heterostructure extracted from the regions indicated by the corresponding colored circles in (**a**) as a function of laser wavelength. $I_R$ and $I_S$ represent the photocurrents generated in the $ReSe_2$ and $SnS_2$ regions, respectively. **e** Schematic diagram showing the trap states induced photogating effect. $\phi_{B1}$ and $\phi_{B2}$ are the built-in potentials between Au and $ReSe_2$ and $SnS_2$, respectively; $V_{bi}$ is the built-in potential at the interface of $ReSe_2$

and $SnS_2$; CB and VB represent conduction band and valence band, respectively; $h\upsilon$ represents the incident photon energy; $E_F$ is the fermi level; $DF_S$ is the defect band with S vacancies in $SnS_2$; $DF_{R1}$, $DF_{R2}$, and $DF_{R3}$ are three defect bands with Re vacancies in $ReSe_2$. Solid and empty circles represent electrons and holes, respectively. Black straight arrows indicate the directions of $I_R$ and and $I_S$; black dotted arrows indicate the excitation of electrons (holes); green curved arrows represent the movement of electrons (holes); double curved arrows indicate the incident photons. **f** Calculated local gate voltage generated by photoexcited carrier trapping at the interface ($\triangle V_g$) as a function of laser wavelengths at different gate voltages.

vacancies, while the $ReSe_2$ has ~2.4 % Re vacancies (Fig. 1c, d and Supplementary Fig. 6). First-principle calculations reveal Re vacancies result in the appearance of three defect bands, labeled $DF_{R1}$ (0.26 eV below bottom of conduction band (CB)), $DR_{R2}$ (0.49 eV below the bottom of CB), and $DF_{R3}$ (0.37 eV above the top of the valence band (VB)). In the case of $SnS_2$ with sulfur vacancies, a single defect band ($DF_S$) emerges, positioned 0.7 eV below the bottom of the CB (Supplementary Fig. 7). Moreover, due to the upward bending of the energy band, hole trapping plays a unique role in the overlapping region as excited holes will move to the overlapped region and electrons move to the nonoverlapped region (Fig. 2e and Supplementary Fig. 8f). Thus, under illumination, holes are excited and move to the overlapping region and some of the holes will be trapped by $DF_{R3}$, resulting a photogating effect, which would have the effect of enhancing the photocurrent. When the light is switched off, the trapped holes remain and sustain the photocurrent, resulting in a memory effect. This is also key to the electrically tunable memory effect, discussed further in the paper. Moreover, the interfacial trap states, such as neutral traps (NT), are generally observed in 2D heterostructures during the stacking process, which also result in photogating and memory effects (Supplementary Fig. 9). Details of the

calculation can be found in Supplementary Note 1. For photodetectors dominated by a photogating effect[29], the photocurrent in the interface ($I_{interface}$) can be written as $I_{interface} = \frac{\partial I_{ds}}{\partial V_g} \triangle V_g$, where $\triangle V_g$ is the local gate voltage generated by photoexcited carrier trapping at the interface[30]. In Fig. 2f, the calculated $\triangle V_g$ is shown to decrease with increasing gate voltage. We would like to stress that the number of holes trapped depends on the built-in potential at $ReSe_2$ ($V_{biR}$). Increasing $V_g$, $V_{biR}$ decreases, resulting in a decrease in the number of trapped holes, which is consistent with the experimental results. The total photocurrent ($I_{ph}$) has three main contributions: the photocurrents generated in the $ReSe_2$ ($I_R$) and $SnS_2$ regions ($I_S$), and the photocurrent generated in the overlapping region ($I_{interface}$) due to the photogating effect. As the Au electrode has a much greater work function in comparison with both $SnS_2$ and $ReSe_2$, the built-in potential between Au and $ReSe_2$ ($\phi_{B1}$) is greater than $V_{biR}$. Thus, $I_S$ and $I_R$ are in opposite directions. Therefore, $I_{ph} = I_S - I_R + I_{interface}$. Moreover, $I_{interface} = \frac{\partial I_{ds}}{\partial V_g} \triangle V_g$ is almost constant when far from the photocurrent peak (15 V in Fig. 2f). Thus, the peak of photocurrent can be explained by $|I_S - I_R|$. Figure 2c plots the $I_R - I_S$ and it does show a peak around 600 nm, which is consistent with the photocurrent of the device. Moreover, the built-in potential of the

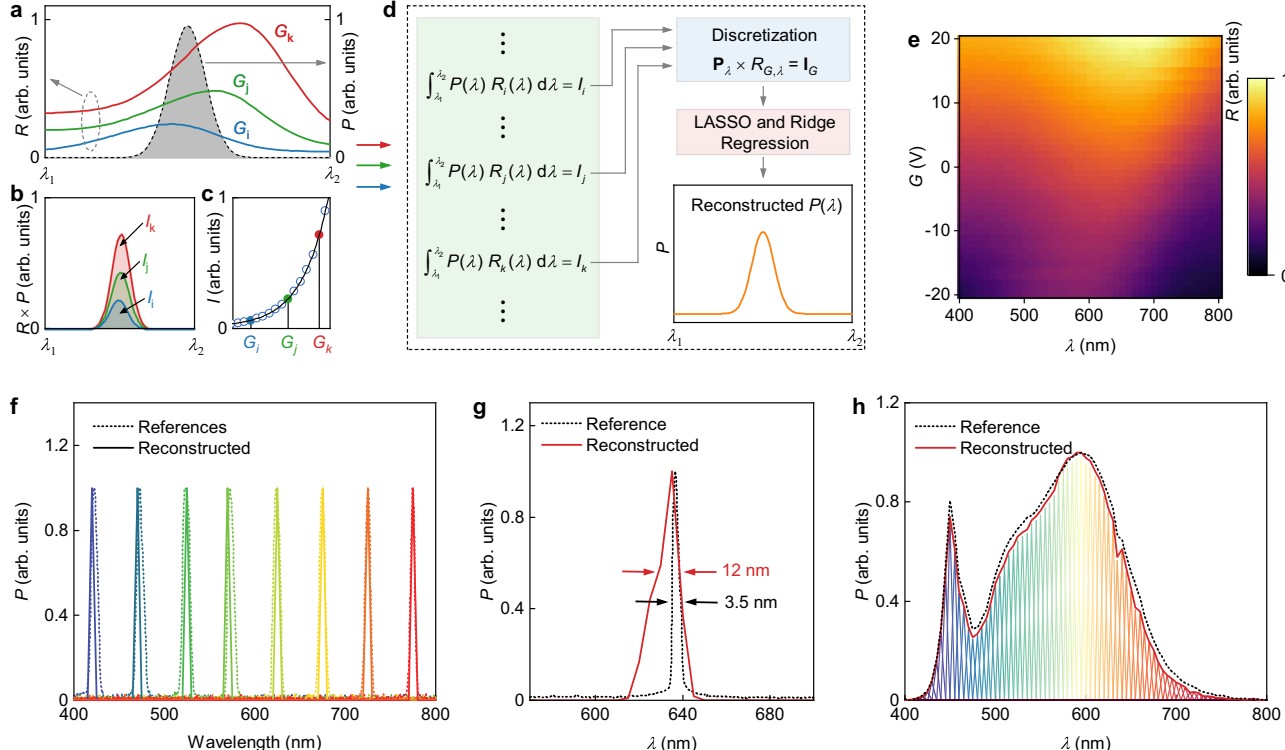

**Fig. 3 | Design of vdW spectrometers. a** Photoresponsivity ($R$) curves at three different gate voltages ($G_i$, $G_j$ and $G_k$, left axis) and a schematic of an unspecified light spectral density ($P$) curve (gray dashed line, right axis). $\lambda_1$ and $\lambda_2$ define the operational spectral range of the spectrometer. **b** The photoresponse spectrum ($R \times P$) of the device under unspecified light illumination source at different gate voltages, $G_i$ (blue), $G_j$ (green) and $G_k$ (red). **c** The integrated photocurrent ($I$) as a function of gate voltage. Empty circles represent some typical data points in the photocurrent curve. **d** Spectral reconstruction process of our spectrometer.

**e** Photoresponsivity of the device as a function of gate voltage ($G$) and light wavelength ($\lambda$). **f** Quasi-monochromatic spectra (FWHM: ~12 nm) reconstructed with our spectrometer (solid curve) and corresponding reference spectra measured using a commercial spectrometer (dashed curve). **g** The reconstructed 635 nm laser spectrum (FWHM: ~12 nm) and the reference spectrum (FWHM: ~3.5 nm) measured using a commercial spectrometer. **h** Reconstructed broadband spectrum of a white flashlight from a mobile phone and the reference spectrum.

SnS$_2$ side decreases the most with increasing $V_g$ and that in the ReSe$_2$ side remains quite stable (Supplementary Fig. 8e). In other words, the photocurrent decreases much faster with increasing $V_g$ (Supplementary Note 2). Thus, the peak position of the photocurrent moves to longer wavelength with increasing $V_g$, which is consistent with the experimental observation (Fig. 2f).

## Spectrum reconstruction

The fabricated SnS$_2$/ReSe$_2$ heterostructure exhibits a gate-tunable spectral response with high responsivity and detectivity across a wide range of wavelengths, making it suitable for use as a spectrometer[1]. The reconstruction process is briefly outlined in Fig. 3 (see Supplementary Fig. 10 for a detailed workflow diagram). In the non-saturated region (linear region, as shown in Fig. 1g) at a given gate voltage (denoted as $G_i$), the photocurrent ($I_i$) can be calculated as an integral of the product of the incident light power density ($P(\lambda)$) and the responsivity ($R_i(\lambda)$) over the entire spectral range of interest from $\lambda_1$ to $\lambda_2$:

$$\int_{\lambda_1}^{\lambda_2} P(\lambda)R_i(\lambda) = I_i \ (i = 1,2,3\ldots,n) \tag{1}$$

where $\lambda_1$ and $\lambda_2$ define the operational spectral range of the spectrometer and index $i$ implies different gate voltage values. Here, the unknown spectrum to be determined is denoted by $P(\lambda)$ and is schematically illustrated by an arbitrary narrowband emission spectrum, represented by the dashed line curve in Fig. 3a. Additionally, three representative responsivity curves corresponding to the three different gate voltages ($G_i$, $G_j$ and $G_k$) are sketched in Fig. 3a. Using Eq. (1), we

can calculate the photocurrent of the device for a given gate voltage, determined by the integral of the shaded area in Fig. 3b. Figure 3c shows the measured photocurrents at all different gate voltages, where $G_i$, $G_j$ and $G_k$ are highlighted in different colors. By varying the gate voltage from $G_1$ to $G_n$, we obtain $n$ integral equations that consider small windows of wavelengths in (1). These equations can be discretized and grouped into a matrix equation:

$$\begin{pmatrix} P_{\lambda_1} & P_{\lambda_2} & \cdots & P_{\lambda_n} \end{pmatrix} \begin{pmatrix} R_{G_1,\lambda_1} & R_{G_2,\lambda_1} & \cdots & R_{G_n,\lambda_1} \\ R_{G_1,\lambda_2} & R_{G_2,\lambda_2} & \cdots & R_{G_n,\lambda_2} \\ \vdots & \vdots & \ddots & \vdots \\ R_{G_1,\lambda_n} & R_{G_2,\lambda_n} & \cdots & R_{G_n,\lambda_n} \end{pmatrix} = \begin{pmatrix} I_{G_1} & I_{G_2} & \cdots & I_{G_n} \end{pmatrix} \tag{2}$$

or $\mathbf{P}_\lambda \times \mathbf{R}_{G,\lambda} = \mathbf{I}_G$ in a more compact form. To enable accurate reconstruction of an unknown spectrum using the SnS$_2$/ReSe$_2$ heterostructure, we employed prior robust principal component analysis (RPCA) on the responsivity matrix $R_{G,\lambda}$ to stabilize the solution against various types of noise and errors. Subsequently, we used a regression model with elastic net regularization to solve the resulting matrix Eq. (2) by minimizing the sum of the squared errors loss function, with the added penalty term being a combination of $\ell_1$ and $\ell_2$ (Lasso and Ridge) norms. Details are provided in Methods and Supplementary Figs. 10 and 11.

To determine the responsivity matrix $R_{G,\lambda}$, we utilized a tunable light source with a wavelength ranging from 400 to 800 nm and 81 sampling points, measured at 81 different voltages from −20 V to 20 V. The resulting 81 × 81 $R_{G,\lambda}$ matrix is shown in Fig. 3e, exhibiting a tunable

response within the visible and near-infrared spectrum range, enabling broadband spectrum reconstruction. To demonstrate the SnS$_2$/ReSe$_2$ heterostructure's capability for reconstructing different types of spectra, we measured a series of quasi-monochromatic spectra in the range of 400 ~ 800 nm with our single-heterostructure spectrometer, which agree well with reference spectra acquired using a commercial spectrometer, verifying the broad spectral range of our device (Fig. 3f). We also demonstrate the reconstruction of 532 nm and 635 nm laser spectra in Supplementary Fig. 12 and Fig. 3g, respectively. Note, while recording the corresponding current values, we continuously illuminated the device with light for 1 min to stabilize the photocurrent, and then swept the gate voltage from -20 V to 20 V (sweep rate 1 V/s). Thus, the time required to measure a single complete spectrum is approximately 100 s, which can be reduced by decreasing the illuminating time, increasing the sweep rate, or/and using compressive sensing techniques. The full-width at half-maximum (FWHM) of the reconstructed laser spectrum's was about 12 nm, with an optimal resolution of 11.3 nm according to the Nyquist–Shannon sampling theorem[31]. Note, although the Shannon-Nyquist theorem is primarily concerned with the sampling and reconstruction of time-dependent signals, in this work, this theorem is extended to the spatial frequency content of the signal and applied to the wavelength sampling. To further showcase device's ability for broadband spectrum reconstruction, we measured the spectrum of a white flashlight from a mobile phone, as shown in Fig. 3h, revealing excellent agreement between the reconstructed and measured reference spectra. We also used the lowest irradiance suppliable by the apparatus available to us (23 μW/cm$^2$) and a faithful reconstruction was produced with such low power density (Supplementary Fig. 13). We measured around 10 devices, and all showed essentially similar capabilities to reconstruct visible light spectra. An imaging array was also fabricated, using chemical vapor deposition (CVD) grown SnS$_2$ and ReSe$_2$ (Supplementary Fig. 14). Although all the devices worked properly, further optimization is needed to improve their performance.

## Compressed sensing

Improving the resolution of a spectrometer is a key challenge in developing advanced spectroscopy techniques. The resolution of a spectrometer is limited by the number of sampling points, as previously discussed. However, the number of sampling points cannot be increased indefinitely due to the numerical properties of the responsivity matrix $R_{G,\lambda}$. To overcome this limitation and achieve a higher resolution[32,33], we employ compressed sensing. For this, we utilized the singular value decomposition (SVD) technique, a well-known compression technique to achieve full spectrum reconstruction from surprisingly few $p$ measurements of $I_G$ as described in the Methods section. In our case, a singular value ($\sigma_i$) represents the contribution of each feature of $R_{G,\lambda}$. To determine the optimal number of sensors $p$ or, alternatively, distinct gate voltage values for measurements, we followed methodology similar to that used by ref. 34. through QR decomposition and column pivoting. By gradually decreasing the $p$ value from 81 to 9 and comparing the mean square error (MSE $= \frac{1}{n} \sum_{i=1}^{n} (I_G - \widetilde{I}_G)^2$), where $\widetilde{I}_G$ represent the reconstructed photocurrent from the $p$ sensors, $I_G$ is the original photocurrent curve, and $n = 81$. Figure 4a shows the singular values $\sigma_r$ and the optimal modal is noted to occur for this system at $r = 9$, capturing 99% of the main features of $R_{G,\lambda}$ ($\frac{\sum_{i=1}^{r} \sigma_i}{\sum_{i=1}^{n} \sigma_i} \times 100$). Figure 4b, c shows measurement matrices for a random and optimized sampling obtained by QR decomposition and column pivoting. The reconstructed photocurrents versus gate voltage $\widetilde{I}_G$ from a random and optimized sampling for $p = 9$ are shown in Fig. 4d. Compressed sensing leads to a more accurate reconstruction of the measured photocurrent than the randomly sampled system. One can also see that the estimated error for

the optimized sampling is much smaller than that for the randomly selected one (Fig. 4e). Our results demonstrate that compressed sensing enables us to reconstruct the photocurrent versus gate voltage from merely 9 selected points (minimum 9 sensors) out of a total of 81, which is around 12% of our original sampling. The optimized sampling corresponds to gate voltages located where the first derivative d$I$/d$V_g$ shows the largest variation (Fig. 4b, c). To ensure accurate compressed sensing, as shown in Fig. 4f, we reconstructed our spectrum using random and optimized sampling with elastic net regression as described in the Methods section. We observed that the spectrum reconstructed from the optimized sampling agrees well with the measured reference spectrum, whereas the reconstruction from random sampling shows an erroneous spectrum. Furthermore, we demonstrate that we can reconstruct the spectrum of a red LED with only 9 points, with very little difference compared to the reconstructed spectrum with $n = 81$ and the reference spectrum (Fig. 4g).

We compared our 2D heterostructure spectrometer with other miniaturized spectrometers in terms of their footprints, spectral range, and resolution, as shown in Fig. 4h (see Supplementary Table 1 for details). Compared to separate optical elements, 2D materials-based spectrometers offer distinct advantages in terms of footprint miniaturization[7,11,15]. Our device has a small footprint of 19 μm and an enhanced spectral resolution of 5 nm, making it comparable to state-of-the-art visible and near-infrared spectrometers. Here, the maximum length along any axis of the heterostructure is considered to be the device footprint. The device has an area of ~220 μm$^2$. Furthermore, the footprint of a 2D materials-based device can be scaled down to sub-micrometer dimensions, and the resolution can be further improved by increasing wavelength sampling points. Our device also has an advantage in processing speed over other computational spectrometers due to the compressed sensing techniques employed. While the spectrum reconstruction capability of our device relies on the nature of SnS$_2$ and ReSe$_2$ with trap states, this approach can be extended to other 2D materials by intentionally introducing trap states[34] to expand the spectral range.

## Spectral imaging and image memory demonstration

To demonstrate the potential application of our single heterostructure spectrometer for spectral imaging, we designed a spatial scanning system for the device (Fig. 5a). A broadband light source was used to project an image consisting of red, green, blue, and yellow colors, which could be scanned across the $x$-$y$ plane on the centimeter scale. The reflected light was focused by a lens onto the device, and we measured the photocurrents of each mapping step. The reflected spectrum was then reconstructed using the method discussed above. The reconstructed spectra at two positions indicated in Fig. 5a are shown in Fig. 5b, which are in good agreement with the corresponding reference spectra. All the measured photocurrents were grouped into a three-dimensional $I(x,y,G)$ data cube (Fig. 5c, e), which was then converted to a spectral data cube $P(x,y,\lambda)$ by the reconstruction algorithm (Fig. 5d, f). By applying standardized color-matching functions to the spectral cube, we were able to form a pseudocolored image (Fig. 5h) with an accurately recreated colored "BIT" pattern, which was consistent with the original image (Fig. 5g). In this configuration, the image resolution is defined by the mapping step, which was set to around 1 cm. By fabricating device arrays, we could produce large-scale imaging systems with the single-heterostructure device possessing high spatial resolution at the micrometer or nanometer scale[35]. Our results demonstrate the potential of our single heterostructure spectrometer in spectral imaging, with the ability to accurately reconstruct the reflected spectrum of a colored image. The device's high spatial resolution at the micrometer or nanometer scale[15], when integrated into device arrays, opens up possibilities for large-scale imaging systems.

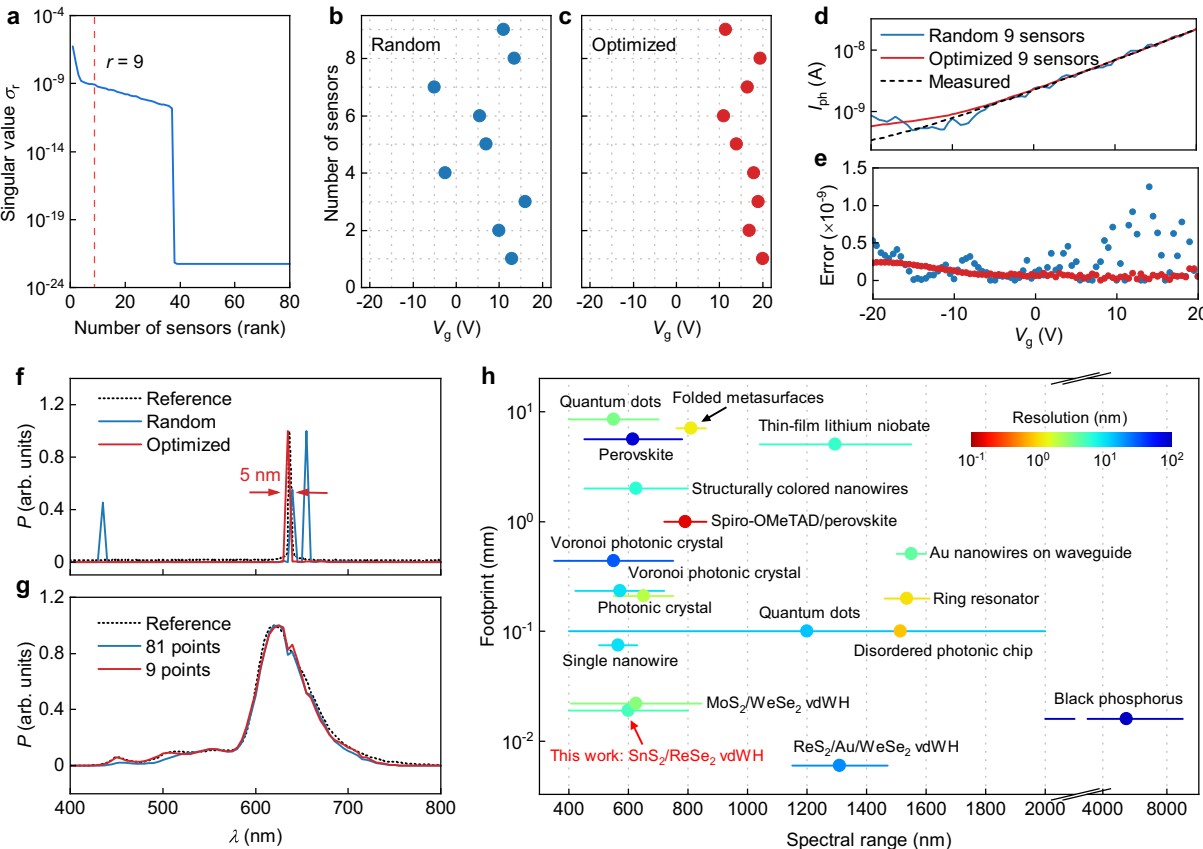

**Fig. 4 | vdW spectrometers improved with compressed sensing. a** Singular values $\sigma_r$ and the optimal rank truncation threshold at $r = 9$ (indicated by the vertical dashed line) for the tailored $\Psi_r$ basis, extracted from the responsivity matrix $R_{G,\lambda}$, where $\gamma$ is the noise magnitude determined by using the median singular value. **b, c** Measurement matrices $C \in \mathbb{R}^{p \times n}$, where $p = 9$ and $n = 81$ for random and optimized sampling, respectively. **d** Reconstructed photocurrent versus gate voltage for random sampling (blue line) and optimized sampling (red line), where the (black dashed line is the measured photocurrent. **e** Discretized error distribution defined by $I_{reconstructed} - I_{measured}$ for the random (blue points) and optimized sampling (red points). **f** Spectral reconstruction for a laser excitation source at 632 nm for random and optimized measurement matrices. **g** Spectral

reconstructions for the optimal sampling through compressed sensing with $p = 9$ compared to the reconstructed spectrum with 81 points, and the reference spectrum. **h** Comparison between our work and other reported miniaturized spectrometers (MoS2/WSe2 vdWH (van der Waals heterostructure)[15], ReS2/Au/WSe2 vdWH[11], black phosphorus[7], single nanowire[14], quantum dots[5,12], photonic crystal[10], structurally colored nanowires[18], interferometer[4], folded metasurfaces[17], perovskite[54], spiro-OMeTAD/perovskite[55], Voronoi photonic crystal[56], thin-film lithium niobate[9], disordered photonic chip[8], holographic[57] and ring resonator[58]). The horizontal lines represent the operational spectral range. Color variation indicates the difference in resolution. The detailed data can be found in Supplementary Table 1.

We have demonstrated the light detection, spectrum reconstruction, and spectral imaging capabilities of our single-heterostructure device, which can also be achieved in other systems such as quantum dots[5,12], nanowires[14,18], black phosphorus[7] and MoS2/WSe2 heterojunctions[15]. However, our SnS2/ReSe2 device has a unique feature in producing a gate-tunable photocurrent via trap states at the interface, offering a tantalizing route for the design of image memory[36,37]. In Fig. 6a, we show the time-dependent variation of current in the device when applying laser illumination and gate pulses. After being illuminated by a 635 nm laser, the current immediately increases sharply and then gradually approaches a set value, indicating a long-term positive persistent photoconductivity (PPC) effect in our device attributed to trap states at the interface[38,39] (see Supplementary Note 1 for details). To further investigate the time evolution of the PPC as well as the switching action, we used laser pulses (pulse-width ~0.1 s), instead of continuous laser illumination, to excite the photocurrent in the device. As expected, after applying the laser pulses, a large quantity of carriers is produced in the device, leading to a rise in the current (Fig. 6b). The carriers are easily retained beyond the timeframe of minutes in dark conditions, completing a program operation. This state is defined as the 'optical program state'. These carriers can be eliminated by applying a positive gate pulse, that is, an

erase operation, setting the current back to its initial value, which is defined as the 'electrical erase state'. Figure 6c shows the retention characteristic of the program and the erase state. It is found that the programming and erasing current are clearly retained after $10^4$ s, though a slight decline appears to occur for $t < 10$ s, after which the current stabilizes (Supplementary Fig. 15). An endurance test of repeated programming and erasing operation was also performed, of which the results are shown in Fig. 6d. It can be observed that both the program and the erase states remain stable even after 1000 cycles. However, such a strong memory effect indicates a slow response speed in our detector. This problem can be simply solved by changing the detection mode from the 'memory mode' to the 'fast mode' (Fig. 6e, f). At $V_{ds} = 0$, our device shows a remarkable short-circuit current and a fast response speed (response time ~0.43 ms, Supplementary Fig. 16). This characteristic essentially enables our device to adapt to different application scenarios. To demonstrate its imaging memory capacity, we performed a proof-of-concept demonstration using three different lasers and scanning masks with different patterns in Fig. 6g. We show that the photocurrent at each mapping position can be used to reconstruct the incident light spectra and can also be coherently stored in the device for a long period of time (Fig. 6h), confirming the imaging memory capacity of our device.

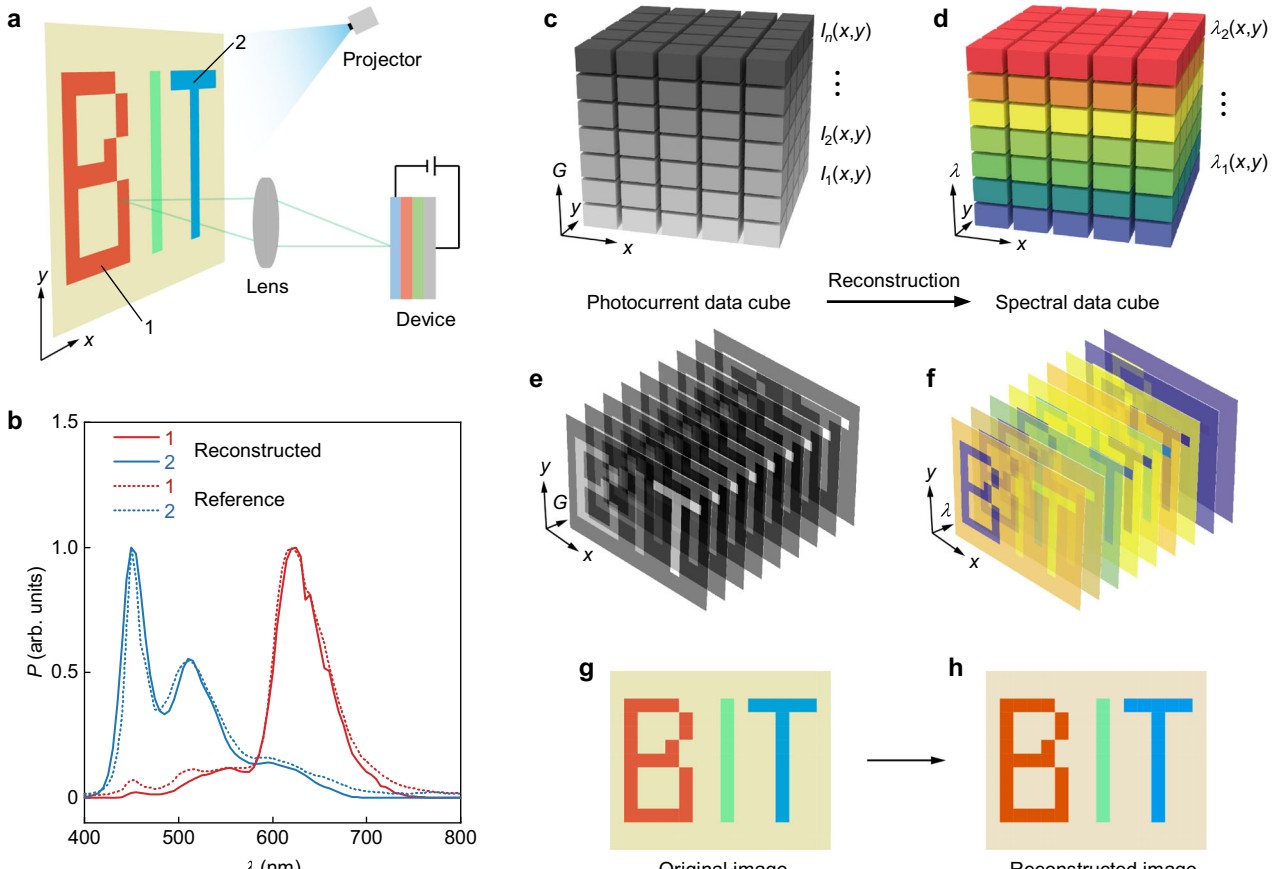

**Fig. 5 | Scanning spectral imaging with vdW spectrometers. a** Schematic of the spectral imaging system with our spectrometer. **b** Reconstructed and reference spectra of points 1 and 2 in (**a**). **c** The initial photocurrent data cube, consisting of measured photocurrent values at each scanned location and gate voltage. **d** The spectral data cube converted from (**c**) using the reconstruction algorithm. A series of (**e**) photocurrent images and (**f**) reconstructed images at each scanned location and selected gate voltages. **g** Original scanned image and (**h**) reconstructed pseudo-colored spectral image.

## Neuromorphic computing spectroscopy

Optical memory effects have been reported by ref. 40. We would like to stress that our device's memory function is due to interface trap states, making it promising for applications in artificial synapses[41], neural networks[38,42], machine vision[42], and imaging processing[43]. To demonstrate the multifunctionality of our compact spectrometer, which not only serves as a storage device but also possesses processing capabilities. We applied periodic light pulses ($P_{in}$ ~ 10 mW/cm², pulse-width ~1 s) to produce an increment in the conductivity, leading to multiple memory states for data storage. Subsequently, by applying periodic gate voltage pulses (amplitude ~5 V, pulse-width ~2 s), we could decrement the conductivity, thus performing erasure the stored states (Fig. 7a). In this context, the increase and decrease in device conductance are termed as potentiation and depression, respectively. Given that our vdW $SnS_2$/$ReSe_2$ device system changes conductance, it can thus be utilized in a neural network and serve as both a neuron and a synapse. Long-term potentiation (LTP) and depression (LTD) cycles were observed, suggesting that our device mimics the function of a synapse with a nonlinear behavior for training the artificial neural network (ANN) (Fig. 7b and Supplementary Fig. 17). Generally, achieving high classification accuracy often relies on the use of artificial synapses that have linear characteristics. However, in our case, we rely on non-linear LTP induced by blue light stimuli and LTD by voltage gate pulses. To attain a high classification accuracy, it is essential to consider the inherent asymmetric non-linearity of our LTP and LTD as well as considering the complete plasticity range represented by the ratio $I_{max}/I_{min}$. To accomplish this, we used an asymmetric nonlinear

relationship to fit the LTP and LTD curves, thereby deriving the nonlinearity (NL) and the weight change of the LTP and LTD, the details are further discussed in Supplementary Note 3. Moreover, the long-term potentiation shape shows a dependency on the incident light, suggesting the possibility of reconstructing the spectrum of incident light with a defined number of memory cells (Fig. 7c). In this study, we simulated a two-layer ANN based on the $SnS_2$/$ReSe_2$ vdW heterostructures synaptic behavior to classify handwritten data on the MNIST data (Fig. 7d). Details of the training process can be found in the Methods section. Figure 7e, f presents the classification accuracies as function of epoch (which represents a single iteration over the complete training dataset) and varying the numbers of synaptic states (ranging from 10 to 40 states). As the synaptic states increase from 10 to 40, the classification accuracy exhibits a gradual improvement, starting at 87% and nearly reaching 90% after just 50 epochs with 40 states. Furthermore, it is noteworthy that employing a greater number of synaptic states results in shorter training times. In Fig. 7f, we have represented the confusion matrix that demonstrates the classification accuracy for each label.

## Discussion

In-sensor storage and processing capabilities are important for next-generation sensing, but also optical sensing is becoming more reliant on computational systems[44]. For such technological changes to become a reality, apart from device reconfigurability, advanced algorithms, would be required to reduce the physical complexity of sensors. Therefore, it is important to develop device concepts that can do more with less,

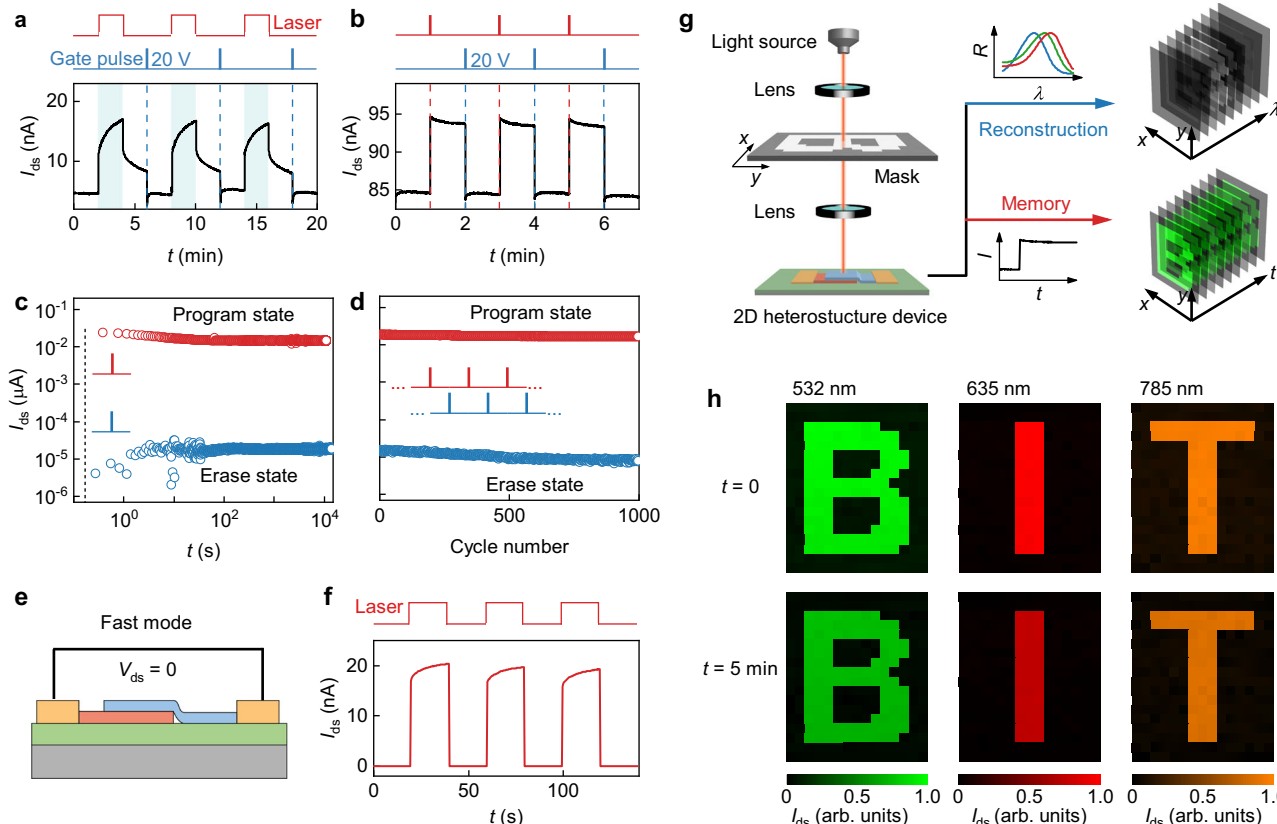

**Fig. 6 | Image memory with vdW spectrometers. a** Time-resolved source-drain current of the device under alternate 635 nm laser illumination and positive gate pulse (pulse-width -0.1 s). The blue shaded area and blue dashed lines indicate the illumination by laser and the application of the gate pulses, respectively. $V_{ds} = 1$ V, $V_g = 20$ V, $P_{in} = 0.2$ mW/cm$^2$. **b** Time-resolved $I_{ds}$ of the device under alternate 635 nm laser pulses and positive gate pulses (pulse-widths are both -0.1 s). The red and blue dashed lines indicate the application of the laser pulses and gate pulses, respectively. $V_{ds} = 1$ V, $V_g = 40$ V, $P_{in} = 0.5$ mW/cm$^2$. **c** Retention characteristic of the optoelectronic memory device after applying the program light pulse (red circles, $P_{in} = 12$ mW/cm$^2$, pulse-width -0.1 s) and erase positive gate pulse (blue circles, amplitude 20 V, pulse-width -0.1 s). The current ratio of program/erase state reaches 10$^3$ at $V_{ds} = 1$ V and $V_g = 0$. The dotted vertical line represents the time when

the laser pulse and gate pulse were applied. **d** Endurance performance test of the optoelectronic device executed with alternate light pulses (red circles, $P_{in} = 12$ mW/cm$^2$, pulse-width -0.1 s) and gate pulses (blue circles, amplitude 20 V, pulse-width -0.1 s). $V_{ds} = 1$ V, $V_g = 0$. **e** Schematic representation of the 'fast mode' of the device at $V_{ds} = 0$. **f** Time resolved current curve of the device under periodic 635 nm laser illumination ($P_{in} = 50$ mW/cm$^2$) at $V_{ds} = 0$ and $V_g = 0$, showing a fast response and response time measurement of the 'fast mode', showing rise and decay times both of 0.43 ms. **g** Schematic of configuration and principle of the imaging memory system of our device. **h** Scanning photocurrent images of three different patterns under three different lasers. The upper panel shows the images at the moment the laser pulses is on ($t = 0$), and the lower panel shows the images at 5 min after the lasers were off ($t = 5$ min).

with designs that can enable direct multiplexing of sensing and computing functions[45]. Here, we showed that a single SnS$_2$/ReSe$_2$ van der Waals heterostructure, can provide photodetection, spectrum reconstruction, spectral imaging, and long-term image memory capabilities. By applying light pulses and gate voltage pulses, our miniaturized spectrometer can realize image-encoding and -classification functions based on artificial neural networks. Although, our approach can potentially overcome the limitations of present single-device systems and pave the way for the design of single-detector computational spectrometers beyond von Neumann architectures. This breakthrough could possibly impact the field of optoelectronics, with applications ranging from artificial synapses, neural networks, machine vision, and imaging processing, to spectroscopy and analytical chemistry.

## Methods

### Device fabrication and characterization

For the fabrication of the vdW heterostructures, a highly n-doped silicon wafer with a 300-nm-thick SiO$_2$ layer was used as the substrate. Heterostructures were fabricated via a standard dry transfer method[46]. Two types of multilayer vdW materials, ReSe$_2$ and SnS$_2$, were obtained by mechanical exfoliation onto PDMS stamps with scotch tape. Then, they were released onto the prepared

substrate to form vertically stacked vdW heterostructures. The heterostructures were annealed in a furnace at 150 °C for 2 h under an Ar (200 sccm) atmosphere to achieve better contact. For the patterning of the electrodes, we utilized standard photolithography and high-vacuum electron-beam evaporation. The thickness of the deposited Ti/Au electrodes was 10/40 nm, respectively. Raman spectra were acquired with a Bruker Senterra confocal spectrometer with an excitation wavelength of 532 nm to identify the properties of SnS$_2$ and ReSe$_2$ as well as their heterostructures. Atomic force microscopy characterization was performed with a Bruker MM8 system and HRTEM characterization was performed using a JEOL JEM-2100 F with a probe size of less than 0.5 nm at a working voltage of 200 kV.

### Reconstruction process

Before spectrum reconstruction, the spectral responsivity matrix $R_{G\lambda}$ was decomposed by RPCA (robust principal component analysis) into a structured low−rank matrix $\widetilde{R}_{G\lambda}$ and a sparse matrix $S_{G\lambda}$, thus discarding the outliers and noise as shown in Supplementary Figs. 10 and 11:

$$R_{G\lambda} = \widetilde{R}_{G\lambda} + S_{G\lambda} \tag{3}$$

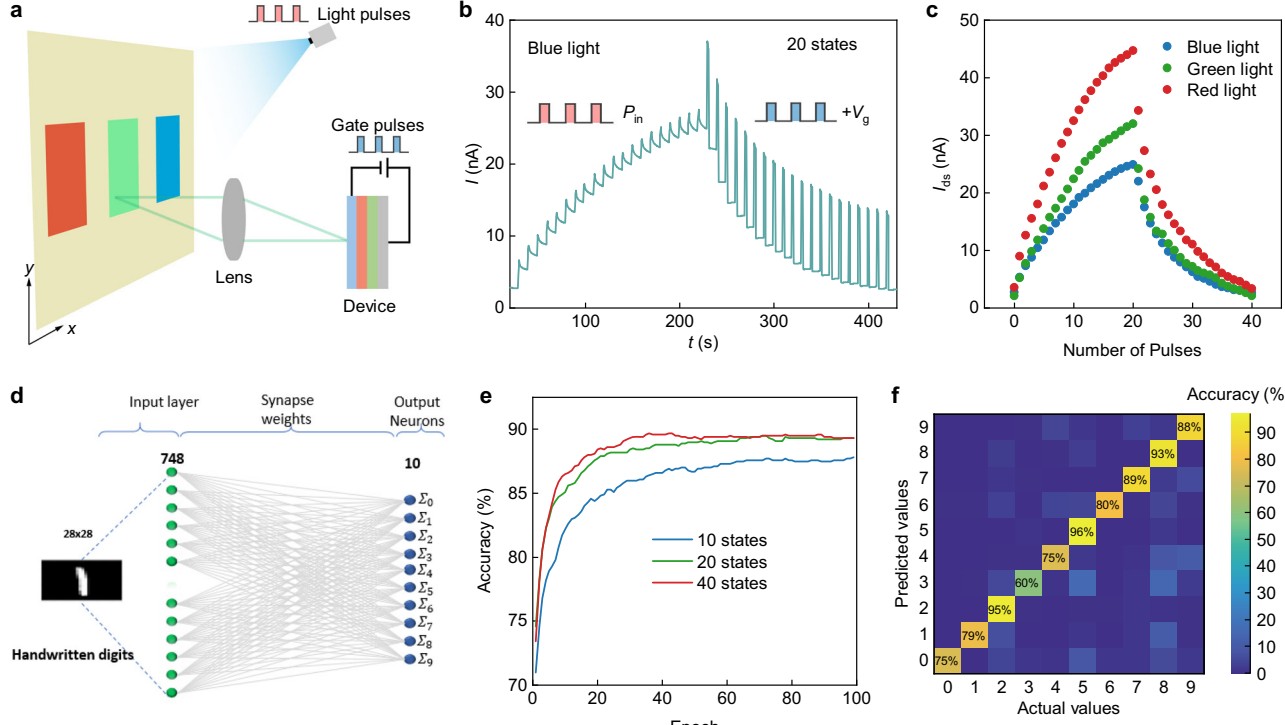

**Fig. 7 | Neuromorphic computing with vdW spectrometers. a** Schematic of the spectral training system for the deep neural networks. A broadband light source, controlled by a signal generator to produce light pulses, was used to project red, green and blue light. The reflected light was focused by a lens onto the device, and we measured the photocurrents of each color. **b** 40 potentiation and depression states under the control of light blue light pulses ($P_{in}$ ∼ 10 mW/cm², pulse-width ∼1 s) and gate pulses (amplitude ∼5 V, pulse-width ∼2 s). **c** Currents of the device under different light illumination as a function of number of pulses. **d** Two-layer artificial neural network for classifying handwritten digits. **e** Neuromorphic classification accuracies as a function of number of iterations for 10, 20 and 40 different synaptic states based on the device. **f** Confusion matrix also known as errors matrix, is used to evaluate the performance of our classifier. The correct classifications are on the diagonal of the matrix, whereas the off diagonal represents the incorrect classifications during the validation test.

where the principal components of $\widetilde{R}_{G,\lambda}$ are robust to the outliers and the noise in $S_{V_G,\lambda}$ matrix. We observe that the $S_{V_G,\lambda}$ sparse matrix is one to two orders of magnitude lower than the $\widetilde{R}_{G,\lambda}$ spectral matrix, and the original spectral matrix $R_{G,\lambda}$. In order to reconstruct the unknown spectrum, we use the elastic net regression that minimizes the sum of squared errors by applying a penalty to these coefficients which is a combination of $\ell_1$ and $\ell_2$ (Lasso and Ridge) approaches. From Eq. (4), we can see that if $\alpha$ is equal to 1, elastic net regression is the same as Lasso regression. As $\alpha$ decreases toward 0, it approaches the Ridge regression. In our reconstruction, we observe a small $\alpha$ (0.01), meaning that we are close to Tikhonov regularization. However, we observe that adding a $\ell_1$ penalization term provides us with a slightly better reconstruction than the Ridge regression alone:

$$\min \left\| \hat{R} \cdot \hat{\varphi} \cdot \vec{\beta} - I \right\|_2^2 + \lambda \left( \underbrace{\frac{1-\alpha}{2} \left\| \vec{\beta} \right\|_2^2}_{\ell_2 \text{ regularization}} + \underbrace{\alpha \left\| \vec{\beta} \right\|_1}_{\ell_1 \text{ regularization}} \right), \quad (4)$$

where $\lambda$ is the amount of penalization, which is chosen by cross-validation.

**Compressed sensing**

The photocurrent values as function of gate voltage $\mathbf{I}_G \in \mathbb{R}^n$ can be represented as a sparse vector $\mathbf{s} \in \mathbb{R}^n$ in a transform basis $\Psi \in \mathbb{R}^{n \times n}$:

$$\mathbf{I}_G = \Psi \mathbf{s} \quad (5)$$

Using compressed sensing, it is possible to effectively achieve full spectrum reconstruction from surprisingly few $p$ measurements

instead of directly measuring $\mathbf{I}_G \in \mathbb{R}^n$ (which requires $n$ measurements). These few $p$ measurements, denoted as $\mathbf{I}_{Gp} \in \mathbb{R}^p$, can then be used to solve the non-zero elements of the transformed sparse vector $\mathbf{s}$. The relationship between the measurements $\mathbf{I}_{Gp} \in \mathbb{R}^p$ and the compressible signal $\mathbf{I}_{Gn} \in \mathbb{R}^n$ is given by the Eq. (6):

$$\mathbf{I}_{Gp} = C\mathbf{I}_G \quad (6)$$

Where, the measurement matrix $C \in \mathbb{R}^{p \times n}$ represents a set of $p$ (sensors) measurements at specific gate voltages.

By combining Eqs. (5) and (6), we obtain the relationship between the compressed measurement $\mathbf{I}_{Gp}$ and the sparse vector $\mathbf{s}$:

$$\mathbf{I}_{Gp} = C\Psi \mathbf{s} \quad (7)$$

In this work, a singular value decomposition technique[47–49] was used to achieve a highly transformative basis. To do this, we first extracted the low-dimensional pattern of the spectral responsivity matrix $R_{G,\lambda}$ through singular value decomposition, leading to a tailored library $\Psi_r \in \mathbb{R}^{n \times r}$, containing the first $r$ columns of $\Psi$ (features or eigenmodes):

$$R_{G,\lambda} = \Psi \Sigma V^T = \Psi_r \Sigma_r V_r^T \quad (8)$$

The diagonal elements of the matrix $\Sigma_r$ are the singular values ($\sigma_i$) of $R_{G,\lambda}$. The rank $r$ is chosen by optimal hard thresholding to capture the main features and to avoid residual noise in the data[50].

Based on the work of ref. 51, in order to find the optimum measurement matrix $C$, the best possible full reconstruction $\tilde{I}_G$ ($n = 81$) is performed by seeking the row of $\Psi_r$ corresponding to the best

location in the $G$ row space of $\Psi_r$ that optimally condition the inversion of $C\Psi_r$. To select the optimum row in the $\Psi_r$ tailored basis, we used reduced matrix QR factorization with column pivoting, enabling the decomposition of $\Psi_r$ into a unitary matrix $Q$, upper-triangular matrix $R$ and a column of permutation matrix $C$ such that $\Psi_r^T C^T = QR$, when the number of $p$ sensors is equal to rank $r$. In the case of the oversampled ($p > r$), where the number of $p$ exceeds the number of features (eigenmodes) used in the reconstruction[52,53], we use the following relationship $\Psi_r \Psi_r^T C^T = QR$. After knowing $\Psi_r$ and the $C$ matrix, we are able to approximate the sparse vector s with the Moore–Penrose pseudo-inverse $\mathbf{s} = \Theta^\dagger \mathbf{I}_{Gp} = (C\Psi_r)^\dagger \mathbf{I}_{Gp}$ for $p > r$ or the inverse $\mathbf{s} = \Theta^{-1}\mathbf{I}_{Gp} = (C\Psi_r)^{-1}\mathbf{I}_{Gp}$ for $p = r$. Finally, when the sparse vector s is known, we can reconstruct the photocurrent $\tilde{I}_G$ from the Eq. (5).

## Neuromorphic computing

To demonstrate the neuromorphic computing capabilities based on our vdW $SnS_2/ReSe_2$ devices, which function as both memory and synapses, we employed a simple neural network for a multiclass classification problem. As depicted in Fig. 7d, the neural network is composed of two layers: an input layer with a dimension of 784 (representing each pixel in the $28 \times 28$ images), and an output layer with 10 neurons, each dedicated to recognizing a specific label ($f_{m,n}$). We trained and tested our neural network by using the MNIST dataset, which contains handwritten digits images (0, 1,…,9).

The training dataset consists of 60,000 images, while the testing dataset contains 10,000 images. During the training process, each input neuron in the input layer receives the corresponding pixel value from the image, assigns it to input vector ($\mathbf{X}_i$), and converts it into 10 outputs values using the linear relation represented as $\Sigma_n = \sum_{i=1}^{784} \mathbf{X}_i W_{i,n}$, where $W_{i,n}$ denotes the synaptic weight matrix. Initially, the weight matrix was randomly initialized based on our vdW $SnS_2/ReSe_2$ synaptic states. To determine the direction of weight updates, the output values $\Sigma_n$ were transformed by using a Softmax activation function ($\sigma(\Sigma)_i = \frac{e^{\Sigma_i}}{\Sigma_{j=1}^{K} e^{\Sigma_j}}$), resulting in a probability distribution denoted as ($O_{m,n}$). When we compare the label value ($f_{m,n}$) of each image 'm' with the probability distribution, it allows us to calculate the delta value $\delta_{m,n} = O_{m,n} - f_{m,n}$). If $\delta_{m,n}$ is greater than 0, the synaptic weight ($w$) is increased; otherwise, the synaptic weight is decreased. The magnitude of the weight changes ($\triangle w$) is determined by the fitting formulas (Supplementary Note 3).

## Data availability

The data that support the findings of this study are available from the corresponding author upon request. Source data are provided in this paper. Source data are provided with this paper.

## Code availability

The code for the simulations are available from the corresponding author upon request. The Code for spectrum reconstruction and neuromorphic computing is provided with the Source Data.

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

## Acknowledgements
This work was supported by the National Natural Science Foundation of China (No. 62374017, 61874010) and the Science and Technology Innovation Program for Creative Talents in Beijing Institute of Technology (No. 2017CX01006). C.-R.C. thanks the support of the National Science and Technology Council Taiwan (R.O.C.), under grant no. NSTC 112-2112-M-033-009.

## Author contributions
H.C.W. designed the experiment. G.W. fabricated the devices and carried out the electrical measurements. M.A. and M.Z. performed the spectrum reconstruction. J.C. and M.C. conducted the TEM characterization. C.Ó.C., K.M.H., C.R.C., and I.V.S. analyzed the data. H.C.W. and G.W. wrote the article. All authors discussed the results and commented on the manuscript.

## Competing interests
The authors declare no competing interests.
