## [Peer Review File · Nature Communications]

Miniaturized spectrometer with intrinsic long-term image memoryREVIEWER COMMENTS

Reviewer #1 (Remarks to the Author):

Gang Wu et al. present a miniaturized spectrometer based on SnS₂/ReSe₂ tunable junctions. In the heterostructure, trap states on the interface also give rise to memory effect, which might have the potential to enable storage and processing capabilities for the miniaturized spectrometer.

Such a miniaturized spectrometer with storage and processing capabilities sounds intriguing. While I read the manuscript with great interest, I notice the device and concept are still immature in several aspects, which require further work and improvements.

1. A potential highlight of this work is the combination of spectrometer and image memory capabilities into a single device. After reading the manuscript, I feel the two parts, spectroscopy and imaging parts are mostly separated. These two functions/modes are configured by the source-drain and gate voltages to work at different scenarios. Do authors have comments on this? Can authors show more comprehensive demonstrations, in which the two functions are seamlessly incorporated?

2. Similarly, authors claim to go beyond von Neumann architecture. But the demo is about a sensor with memory capability. Though this could be a foundation for further development, the current demo is too preliminary to make the claim. Authors are encouraged to demo its signal processing capabilities if possible.

3. Figure S10 illustrates two working modes, which is pivotal for this paper. Please consider moving the figure to the main text for better visibility and clarity.

4. The authors use the unit micrometer for their reported footprints. It is unclear whether the authors use the maximum length and width or the device area for these measurements. Clarification on this point would be helpful as device area is more commonly used in other papers.

5. Lastly, in-sensor storage and processing capabilities are important for next-generation sensing. Expanding the scope to more generalized sensing concepts in the outlook part will help to justify the motivation, such as those discussed in Nature Machine Intelligence 3, 556–565, (2021) and Science 379, 1103 (2023)

Overall, the topic and claims of this manuscript are intriguing. But authors need to provide more demonstrations to solidify their claims.

Reviewer #2 (Remarks to the Author):

The submitted work presents a miniaturised spectrometer device based on a 2d material heterojunction photodetector for which the spectral response can be tuned via the application of a gate voltage. Its operation has been optimised through compressive sensing, and it shows a persistent photocurrent effect that could prove promising for optical memory-related applications.

In general, I commend the authors on producing what appears to be a highly robust piece of work, where the vast majority of the device's (relevant) functionalities and characteristics have been tested thoroughly, and the data is, on the whole, presented clearly.

My primary concerns with the work are its significance when taken in respect to prior works that I have listed in references [1-3]. Unfortunately, in its current form I do not believe it represents a significant enough advance in miniaturised spectral sensing to warrant publication in Nature Communications. Of course, I would invite the authors to argue against the below points, and, if they believe there is a valid case for novelty, to restructure the paper to emphasise as such.

The design, operation and functionality of the device as a spectrometer and spectral imager is, in essence, identical to those in references [1] and [2] (albeit with different materials), with similar resolution and footprint to that in [1]. As far as I can tell, the most substantive difference in the spectrometer operation is this work's use of compressive sensing to select the optimal gate voltages (though I will add later that this isn't particularly well presented), which shows impressive ingenuity, but I don't believe significantly differentiates it.

The optical memory effect within a 2d material heterojunction is something that has also been reported on before, for instance in reference [3]. While this does offer an added element of interest to this work, it does not confer any extra functionality to the device as a spectrometer. It merely shows two previously demonstrated functions, present in the same device, but essentially independent of one another.

On this note, while I am not an expert in computer architectures, it seems to me that it may be quite a stretch to say that this would lead to devices "beyond von Neumann architectures" when only the storage might be integrated into the detector. The introduction is in fact possibly misleading – by stating that a von Neumann architecture is one where "the spectral sensing storage and processing modules are spatially separated" – while the storage and spectral sensing may be integrated to a degree, it is not clear to me how this device, or an advance on it, would ever be able to perform any of the reconstructive processing in the same unit, which I would have thought is likely the biggest bottleneck in terms of energy consumption.

Further to this more general set of concerns, I have a range of other revisions that I suggest would also need to be made before the manuscript becomes publishable.

1. I note that in Fig. 2 (and later in Fig. 3) there is only a demonstration of reconstruction of a peak around 640 nm. For the authors to claim the operational range that they do (400 – 800 nm), there needs to be a demonstration of comparable resolution across this. I am especially concerned here given the roughly Gaussian nature of the spectral responses (Fig. 2f) implying that there is a "concentration" of calibration data (i.e. varied response information) around 600 nm, and that toward 400 and 800nm the performance may dip significantly. I appreciate there is a "broadband" spectrum in Fig. 2h, but this is not

sufficient to claim comparable operation across that region given the breadth of the spectral features in that test spectrum, and especially given that the intensity tails off before 450nm and beyond 700 nm anyway.

2. What is the dynamic range of the spectrometer? Or, also, what is the lowest irradiance where a faithful reconstruction can be produced? Given the size of the detector itself, I would be concerned that the intensity of light needs to be quite large before an appreciable signal is reproduced. These values should be established to address this.

3. It seems as far as I can tell that there is only one device tested here - what is the device to device repeatability? Even if not good, at the very least I would expect some reference to it. If it is satisfactory, while not essential, a demonstration of a rudimentary snapshot spectral imaging array (even just a 3x3 grid of devices for instance) would increase the importance of the work significantly.

4. While the initial response time is on the order of 10s of milliseconds (at least, in "fast" mode), the PC still increases beyond this (and in "memory" mode, decreases) over the order of seconds. It strikes me therefore that it may be necessary to wait until the photoresponse has plateaued before the data can be collected? And then also this must be repeated at each value of gate voltage? If this is the case I imagine that the data collection time becomes significant – possibly on the order of 10s of seconds or more. Either way, for transparency, the rates of acquisition should at the very least be mentioned in the manuscript, especially if it is notably slow. To me, the current writing gives the impression that one measurement can be taken in a snapshot, rather than needing to scan across the different gate voltages.

5. If there is a persistent photocurrent after each illumination, does that mean that the gate voltage needs to be pulsed back and forth between each value to "reset" the device? Or can it simply be scanned across the range of values? If the former this should be mentioned.

6. The figures (3a-e) and discussion concerning the compressive sensing is poorly explained even for me, and would be extremely unclear for anyone who is not familiar with compressive sensing. In general I would advise completely re-writing this section to make it more accessible, on top of some more specific points below:

a. It is not clear to me (nor earlier in discussion of figure 2) in what sense you are referring to the Nyquist-Shannon theorem, given that we are not "sampling" the spectrum in a traditional manner. Further clarification in the text is necessary.

b. Fig 3a: the one sentence explanation of this figure in the text is unsatisfactory – it is not really clear why you have chosen 9 sensors, given that this doesn't actually seem to be at the inflection point in the plot? In what sense is this "capturing 99% of the energy"? The non-specialised reader will also not know what a "singular value" is.

c. Figs 3b,c: It is extremely unclear what these measurement matrices actually represent and what conclusions should be drawn from them. What does the colour scale signify?

7. Other more minor notes:

a. Fig. 1g: Captioned wrongly.

b. Fig. 2f: It is unclear in what way you are “reconstructing” the photocurrent values here or more importantly, why, given that typically in this field (and within this paper mostly) reconstruction refers to measurement of light spectra.

c. Fig. 3b: No horizontal axis.

d. Fig. 3h: All the works on this plot need to be referenced (i.e. the reference number labelled alongside the description of the class of device in the plot).

e. Fig. 5c: What does the dotted vertical line represent?

References

[1] Deng, W. et al. Electrically tunable two-dimensional heterojunctions for miniaturized near-infrared spectrometers. *Nat Commun* 13, 4627 (2022).

[2] H. H. Yoon et al. Miniaturized spectrometers with a tunable van der Waals junction. *Science* 378, 296–299 (2022).

[3] Xiang, D. et al. Two-dimensional multibit optoelectronic memory with broadband spectrum distinction. *Nat Commun* 9, 2966 (2018).

Reviewer #3 (Remarks to the Author):

The article titled "Miniaturized spectrometer with intrinsic long-term image memory" by Wu G. et al. introduces a miniaturized spectrometer that utilizes a 2D SnS₂/ReSe₂ van der Waals heterostructure. The device is capable of photodetection, spectrum reconstruction, and spectral imaging within the 400–800 nm bandwidth with a spectral resolution of 5 nm. Additionally, the non-volatile storage of photo-generated carriers with long retention characteristics of 10⁴ s is observed in the heterostructure spectrometer due to interfacial trap states. This work presents a straightforward method to achieve a single non-volatile spectrometer by stacking atomic layered materials, which could be of interest to the 2D optoelectronics and image processing communities in Nature Communications. However, several key conclusions in this work lack sufficient support from experimental observations and simulation results. The mechanism behind the non-volatile heterostructure spectrometer's operation remains unclear, and its potential application beyond von Neumann architectures is uncertain. I recommend that the authors address these issues before publishing this work.

1. Previously, single spectrometers based on 1D and 2D materials have been reported, exhibiting both wide bandwidth and high spectral resolution (*Science* 365, 1017–1020 (2019); *Nano Lett.*, 20, 320–328 (2020); *Science* 378, 296–299 (2022); *Nat. Commun.* 13:4627 (2022)). In an attempt to differentiate their work from these reports, the authors assert that their SnS₂/ReSe₂ spectrometer, operating in a non-

volatile manner, could be applied beyond von Neumann architectures. However, the simple demonstration of the imaging memory system in Fig. 5 does not adequately reveal its potential use in non-von Neumann architectures such as artificial neural networks or machine vision. To strengthen the motivation of their work and differentiate their spectrometer from previous reports, the authors should provide additional experimental results and discussions on how their heterostructure spectrometer can be utilized in non-von Neumann architectures.

2. The SnS₂/ReSe₂ device exhibits long-term positive persistent photoconductivity, which is distinct from the fast photoresponse behavior reported in the MoS₂/WSe₂ heterostructure (Science 378, 296–299 (2022)). The authors claim that such non-volatile photoresponse is induced by the interfacial trap states in the SnS₂/ReSe₂ heterostructure. However, it remains unclear how these trap states are introduced in the heterostructure compared to the MoS₂/WSe₂ device. Additionally, it is essential to understand how the trap states affect the dynamic transport behavior of photo-generated carriers in the heterostructure, which is likely more complex than the explanation provided for a single material in Fig. S8. Furthermore, it would be helpful to know if there are any controllable methods for introducing trap states in the SnS₂/ReSe₂ heterostructure, which could consistently induce such non-volatile photoconductivity. To validate their claim of interfacial trap states induced long-term photoconductivity, the authors should provide more explanations and experimental results.

3. Although the trap states induced photogating effect dominates the SnS₂/ReSe₂ heterostructure photodetector, it is still unclear why the device exhibits a peak when the wavelength is changed in the visible range. To clarify the working mechanism of the heterostructure spectrometer, the authors should provide more experimental results and discussions. Additionally, besides the SnS₂/ReSe₂ and MoS₂/WSe₂ heterostructures, it would be interesting to investigate whether similar spectrometers can be achieved by stacking other 2D materials. Furthermore, it is essential to explore whether there are any specific requirements regarding the band offsets in the heterostructures.

4. It would be beneficial to demonstrate laser spectra at other wavelengths ranging from 400 nm to 800 nm in addition to the 635 nm laser spectrum reconstructed and measured using a commercial spectrometer in Fig. 2g. The same applies to the measurements in Fig. 3f, where it would be useful to illustrate more spectra with different wavelengths.

Response to reviewers' comments

Reviewer #1 (Remarks to the Author):

Comment: Gang Wu et al. present a miniaturized spectrometer based on SnS₂/ReSe₂ tunable junctions. In the heterostructure, trap states on the interface also give rise to memory effect, which might have the potential to enable storage and processing capabilities for the miniaturized spectrometer.

Such a miniaturized spectrometer with storage and processing capabilities sounds intriguing. While I read the manuscript with great interest, I notice the device and concept are still immature in several aspects, which require further work and improvements.

A potential highlight of this work is the combination of spectrometer and image memory capabilities into a single device. After reading the manuscript, I feel the two parts, spectroscopy and imaging parts are mostly separated. These two functions/modes are configured by the source-drain and gate voltages to work at different scenarios. Do authors have comments on this? Can authors show more comprehensive demonstrations, in which the two functions are seamlessly incorporated?

Response: Thank you very much for highlighting the weak aspects of the initial submission, i.e. the two functions in our device were not necessarily interconnected. To bridge this gap, according to your suggestion, in the revised version we include additional experiments performed to showcase that our miniaturized spectrometer not only serves as storage but also has the processing capabilities (**Figure R1**). To do that, periodic light pulses ($P_{in} \sim 10 \text{ mW/cm}^2$, pulse-width $\sim 1 \text{ s}$) were applied to the device to produce multiple memory states for data storage, and periodic gate pulses (amplitude $\sim 5 \text{ V}$, pulse-width $\sim 2 \text{ s}$) were then applied to erase the stored data. Moreover, the observed long-term potentiation and depression cycles, suggest that our device mimics the function of a synapse with a nonlinear behavior for training the Artificial Neural Network. Indeed, we obtained a classification accuracy up to 90% for 50 epochs, which opens up possibilities for the development of a multifunctional sensor (camera), which can be used as storage and as neuromorphic computing. In fact, our attempt also suggests that, by utilizing Deep Neural networks, we can potentially reconstruct the spectrum of the incident light with a certain number of memory cells since the long-term potentiation shape show a dependency on incident light.

Figure R1. a) Schematic of the spectral training system for the deep neural networks. A broadband light source, controlled by a signal generator to produce light pulses, was used to project red, green and blue light. The reflected light was focused by a lens onto the device, and we measured the photocurrents of each color. **b)** 40 potentiation and depression states under the control of light blue light pulses ($P_{in} \sim 10 \text{ mW/cm}^2$, pulse-width $\sim 1 \text{ s}$) and gate pulses (amplitude $\sim 5 \text{ V}$, pulse-width $\sim 2 \text{ s}$). Currents of the device under different light illumination as a function of number of pulses. **d)** Two-layer artificial neural network for classifying handwritten digits. **e)** Neuromorphic classification for different 10, 20 and 40 different states based on the device. **f)** Confusion matrix evaluating the performance of our classifier, the correct classifications are on the diagonal of the matrix, whereas the off diagonal represent the incorrect classifications.

These important discussion points have added to revision as well. See Figure 6 in page 17 and from line 11 in page 17 to line 10 in page 18. **“Optical memory effects have been reported by Xiang et al.. We would like to stress that our device's memory function is due to interface trap states, making it promising for applications in artificial synapses, neural networks, machine vision and imaging processing. To showcase that our miniaturized spectrometer not only serves as storage but also has the processing capabilities. Periodic light pulses ($P_{in} \sim 10 \text{ mW/cm}^2$, pulse-width $\sim 1 \text{ s}$) were applied to the device to produce multiple memory states for data storage, and periodic gate pulses (amplitude $\sim 5 \text{ V}$, pulse-width $\sim 2 \text{ s}$) were then applied to erase the stored data (Fig. 6a). Long-term potentiation and depression cycles were observed, suggesting that our device mimics the function of a synapse with a nonlinear behavior for training the Artificial Neural Network (ANN) (Fig. 6b). Moreover, the long-term potentiation shape shows a dependency on incident light, suggesting that the possibility to reconstruct the spectrum of incident light with a defined number of memory cells (Fig. 6c). We simulated**

a two-layer ANN based on the SnS₂/ReSe₂ vdW heterostructures synaptic behavior to classify handwritten data on the MNIST data (Fig. 6d). Details of the simulation can be found in the Methods section. Figs. 6e and 6f show the classification accuracies for different states (from 10 to 40 states). As the number of states increases from 10 to 40, the classification accuracy slightly increases from 87% until it reaches nearly 90% after only 50 epochs for 40 states. Moreover, the greater the number of the states, the shorter the training time.”

Comment: Similarly, authors claim to go beyond von Neumann architecture. But the demo is about a sensor with memory capability. Though this could be a foundation for further development, the current demo is too preliminary to make the claim. Authors are encouraged to demo its signal processing capabilities if possible.

Response: Thank you very much for highlighting this issue. As noted in the above comment, we have performed additional experiments to showcase that our miniaturized spectrometer not only serves as storage but also has signal processing capabilities, such as for neuromorphic computing, which was demonstrated by the recognition of handwritten digits. Thus, our proposed miniaturized spectrometer is not limited to von Neumann architecture.

Comment: Figure S10 illustrates two working modes, which is pivotal for this paper. Please consider moving the figure to the main text for better visibility and clarity.

Response: Thank you for your valuable suggestion. According to your suggestion, we have moved the fast working modes illustration to the main text in the revision (**Figure R2**). Thus, in new Figure 5, we have both working modes. See new Figure 5 in page 14 and from line 14 to line 18 in page 16. **“This problem can be simply solved by changing the detection mode from the ‘memory mode’ to the ‘fast mode’ (Fig. 5e and 5f). At $V_{ds} = 0$, our device shows a remarkable short-circuit current and a fast response speed (response time ~ 0.43 ms, Supplementary Fig. 14). This characteristic essentially enables our device to adapt to different application scenarios. ”**

Figure R2. Figure 5 | Image memory with vdW spectrometers. **a)** Time-resolved source-drain current of the device under alternate 635 nm laser illumination and positive gate pulse (pulse-width ~ 0.1 s). The red shaded area and blue dashed lines indicate the illumination by laser and the application of the gate pulses, respectively. $V_{ds} = 1$ V, $V_g = 20$ V, $P_{in} = 0.2$ mW/cm². **b)** Time-resolved I_{ds} of the device under alternate 635 nm laser pulses and positive gate pulses (pulse-widths are both ~ 0.1 s). The red and blue dashed lines indicate the application of the laser pulses and gate pulses, respectively. $V_{ds} = 1$ V, $V_g = 40$ V, $P_{in} = 0.5$ mW/cm². **c)** Retention characteristic of the optoelectronic memory device after applying the program light pulse (red circles, $P_{in} = 12$ mW/cm², pulse-width ~ 0.1 s) and erase positive gate pulse (blue circles, amplitude 20 V, pulse-width ~ 0.1 s). The current ratio of program/erase state reaches 10^3 at $V_{ds} = 1$ V and $V_g = 0$. **d)** Endurance performance test of the optoelectronic device executed with alternate light pulses (red circles, $P_{in} = 12$ mW/cm², pulse-width ~ 0.1 s) and gate pulses (blue circles, amplitude 20 V, pulse-width ~ 0.1 s). $V_{ds} = 1$ V, $V_g = 0$. **e)** Schematic representation of the 'fast mode' of the device at $V_{ds} = 0$. **f)** Time resolved current curve of the device under periodic 635 nm laser illumination ($P_{in} = 50$ mW/cm²) at $V_{ds} = 0$ and $V_g = 0$, showing a fast response and response time measurement of the 'fast mode', showing a rise time and a decay time of both 0.43 ms. **g)** Schematic of configuration and principle of the imaging memory system of our device. **h)** Scanning photocurrent images of three different patterns under three different lasers. The upper panel shows the images at the moment the laser pulses is on ($t = 0$), and the lower panel shows the images at 5 minutes after the lasers are off ($t = 5$ min).

Comment: The authors use the unit micrometer for their reported footprints. It is unclear whether the authors use the maximum length and width or the device area for these

measurements. Clarification on this point would be helpful as device area is more commonly used in other papers.

Response: Thank you very much for highlighting the issue of ambiguity in the device footprints. In this work, the maximum length along any axis of the heterostructure was 19 μm which is considered to be the device footprint. Furthermore, the device has an area of $\sim 220 \mu\text{m}^2$. We have clarified this point in the revision. See from line 13 to line 15 in page 12. **“Here, the maximum length along any axis of the heterostructure is considered to be the device footprint. The device has an area of $\sim 220 \mu\text{m}^2$.”**

Comment: Lastly, in-sensor storage and processing capabilities are important for next-generation sensing. Expanding the scope to more generalized sensing concepts in the outlook part will help to justify the motivation, such as those discussed in Nature Machine Intelligence 3, 556–565, (2021) and Science 379, 1103 (2023)

Response: Thank you very much for making this good suggestion. According to your suggestion, we added the following sentences to the Discussion section to highlight the importance of in-sensor storage and processing capabilities for next-generation sensing, and the useful references have been cited in revision as well. See Reference 19 and 50 in page 25 and from line 12 to line 17 in page 18. **“In-sensor storage and processing capabilities are important for next-generation sensing, but also optical sensing is becoming more reliant on computational systems. For such technological changes to become a reality, apart from device reconfigurability, advanced algorithms, would be required to reduce the physical complexity of sensors. Therefore, it is important to design new device concepts that can do more with less, with designs that can enable direct multiplexing of sensing and computing functions.”**

Comment: Overall, the topic and claims of this manuscript are intriguing. But authors need to provide more demonstrations to solidify their claims.

Response: Thank you very much for your thoughtful and instructive suggestions. We have made all the corrections you suggested, and hope you will consider publishing our work in Nature Communications.

Reviewer #2 (Remarks to the Author):

Comment: The submitted work presents a miniaturised spectrometer device based on a 2d material heterojunction photodetector for which the spectral response can be tuned via the application of a gate voltage. Its operation has been optimised through compressive sensing, and it shows a persistent photocurrent effect that could prove promising for optical memory-related applications.

In general, I commend the authors on producing what appears to be a highly robust piece of

work, where the vast majority of the device's (relevant) functionalities and characteristics have been tested thoroughly, and the data is, on the whole, presented clearly.

My primary concerns with the work are its significance when taken in respect to prior works that I have listed in references [1-3]. Unfortunately, in its current form I do not believe it represents a significant enough advance in miniaturised spectral sensing to warrant publication in Nature Communications. Of course, I would invite the authors to argue against the below points, and, if they believe there is a valid case for novelty, to restructure the paper to emphasise as such. The design, operation and functionality of the device as a spectrometer and spectral imager is, in essence, identical to those in references [1] and [2] (albeit with different materials), with similar resolution and footprint to that in [1]. As far as I can tell, the most substantive difference in the spectrometer operation is this work's use of compressive sensing to select the optimal gate voltages (though I will add later that this isn't particularly well presented), which shows impressive ingenuity, but I don't believe significantly differentiates it.

Response: We would like to thank you for your valuable comments, and we agree with you that the resolution and footprint of the presented devices are similar to those in references [1] and [2]. However, we would like to point that the mechanisms for spectrum reconstruction are very different from those mentioned works. The mechanisms for spectrum reconstruction in reference [1] and [2] are interlayer excitons and band alignment respectively. In our work, the interface trap states induced a gate-tunable and wavelength-dependent photogating effect, which is responsible for spectrum reconstruction, and is also especially important for the long-time memory effect. In fact, due to the presence of the interface trap states, periodic light pulses can be applied to produce multiple memory states for data storage, and periodic gate pulses could be used to erase the stored data, which sets it apart from the other publications. Moreover, long-term potentiation and depression cycles were observed, which suggests that our device mimics a synapse with a nonlinear behavior for training an Artificial Neural Network. Indeed, we obtained a classification accuracy up to 90% for 50 epochs, which opens up possibilities for the development of a multifunctional sensor (camera), which can be used as storage and as neuromorphic computing. In fact, our attempt also suggests that, by utilizing Deep Neural networks, we can potentially reconstruct the spectrum of the incident light with a certain number of memory cells since the long-term potentiation shape show a dependency on incident light.

These important discussion points have added to revision as well. See Figure 6 in page 17 and from line 11 in page 17 to line 10 in page 18. **“Optical memory effects have been reported by Xiang et al.. We would like to stress that our device's memory function is due to interface trap states, making it promising for applications in artificial synapses, neural networks, machine vision and imaging processing. To showcase that our miniaturized spectrometer not only serves as storage but also has the processing capabilities. Periodic light pulses (P_{in} ~10 mW/cm², pulse-width ~1 s) were applied to the device to**

produce multiple memory states for data storage, and periodic gate pulses (amplitude ~5 V, pulse-width ~2 s) were then applied to erase the stored data (Fig. 6a). Long-term potentiation and depression cycles were observed, suggesting that our device mimics the function of a synapse with a nonlinear behavior for training the Artificial Neural Network (ANN) (Fig. 6b). Moreover, the long-term potentiation shape shows a dependency on incident light, suggesting that the possibility to reconstruct the spectrum of incident light with a defined number of memory cells (Fig. 6c). We simulated a two-layer ANN based on the SnS₂/ReSe₂ vdW heterostructures synaptic behavior to classify handwritten data on the MNIST data (Fig. 6d). Details of the simulation can be found in the Method section. Fig. 6e and 6f show the classification accuracies for different states (from 10 to 40 states). As the number of states increases from 10 to 40, the classification accuracy slightly increases from 87 % until it reaches nearly 90% after only 50 epochs for 40 states. Moreover, the greater the number of the states, the shorter the training time. ”

Comment: The optical memory effect within a 2d material heterojunction is something that has also been reported on before, for instance in reference [3]. While this does offer an added element of interest to this work, it does not confer any extra functionality to the device as a spectrometer. It merely shows two previously demonstrated functions, present in the same device, but essentially independent of one another.

Response: Thank you very much for highlight this apparent shortcoming. As noted in the above comments, we have performed additional experiments to showcase that our miniaturized spectrometer not only has functionality for storage but also has processing capabilities, such as a neuromorphic computing capability, which was demonstrated with the recognition of handwritten digits. In fact, our attempt also suggests that, by utilizing Deep Neural networks, we can potentially reconstruct the spectrum of the incident light with a certain number of memory cells since the long-term potentiation shape show a dependency on the incident light.

Comment: On this note, while I am not an expert in computer architectures, it seems to me that it may be quite a stretch to say that this would lead to devices “beyond von Neumann architectures” when only the storage might be integrated into the detector. The introduction is in fact possibly misleading – by stating that a von Neumann architecture is one where “the spectral sensing storage and processing modules are spatially separated” – while the storage and spectral sensing may be integrated to a degree, it is not clear to me how this device, or an advance on it, would ever be able to perform any of the reconstructive processing in the same unit, which I would have thought is likely the biggest bottleneck in terms of energy consumption.

Response: Thank you very much for raising this interesting point. As mentioned in the above comments, we performed additional experiments to showcase that our miniaturized spectrometer not only functions as memory storage but also has signal processing capabilities, such as a neuromorphic computing capability, which was demonstrated with the recognition of

handwritten digits. It performs the usual tasks ascribed to a synapse (which stores the weighted inputs as pulses) and a neuron (which performs the computation task through potentiation or depreciation) which, in turn, supplies the main building blocks of a neuromorphic computer. A von Neumann machine must include instruction memory, data memory, and an execution unit. The machine first fetches the instruction code from the instruction memory and then executes the instruction to do the required work. In neuromorphic computers, however, tasks are initiated by input data, as shown in **Figure R1d**. Thus, our proposed miniaturized spectrometer does indeed qualify to function beyond the limitations imposed by von Neumann architecture.

Comment: Further to this more general set of concerns, I have a range of other revisions that I suggest would also need to be made before the manuscript becomes publishable.

I note that in Fig. 2 (and later in Fig. 3) there is only a demonstration of reconstruction of a peak around 640 nm. For the authors to claim the operational range that they do (400 – 800 nm), there needs to be a demonstration of comparable resolution across this. I am especially concerned here given the roughly Gaussian nature of the spectral responses (Fig. 2f) implying that there is a “concentration” of calibration data (i.e. varied response information) around 600 nm, and that toward 400 and 800nm the performance may dip significantly. I appreciate there is a “broadband” spectrum in Fig. 2h, but this is not sufficient to claim comparable operation across that region given the breadth of the spectral features in that test spectrum, and especially given that the intensity tails off before 450nm and beyond 700 nm anyway.

Response: Thank you for noting this problem. A number of tests were performed to calibrate and qualify our device. We measured a series of quasi-monochromatic spectra in the range of 400~800 nm with our single-heterostructure spectrometer, which agree well with the reference spectra using a commercial spectrometer, verifying the broad spectral range of our device (**Figure R3**).

Figure R3. Quasi- monochromatic spectra (FWHM: ~12 nm) reconstructed with our

spectrometer (solid curve) and corresponding reference spectra measured using a commercial spectrometer (dashed curve).

This important piece of information has been added to the main text in revision as well. See new Fig. 2f in page 7 and from the last 2 lines in page 8 to line 3 in page 9. **“To demonstrate the SnS₂/ReSe₂ heterostructure’s capability for reconstructing different types of spectra, we measured a series of quasi-monochromatic spectra in the range of 400~800 nm with our single-heterostructure spectrometer, which agree well with reference spectra acquired using a commercial spectrometer, verifying the broad spectral range of our device (Fig. 2f).”**

Comment: What is the dynamic range of the spectrometer? Or, also, what is the lowest irradiance where a faithful reconstruction can be produced? Given the size of the detector itself, I would be concerned that the intensity of light needs to be quite large before an appreciable signal is reproduced. These values should be established to address this.

Response: Thank you for posing these interesting questions. According to your suggestion, we measured the response of the device when illuminated with 635 nm light with different power densities (**Figure R4**). Our results suggest that the response depends linearly on the light power density, which is consistent with earlier measurements. To get the dynamic range or the lowest irradiance, we measured the I - V_g and spectrum with lowest power density inside the group ($23 \mu\text{W}/\text{cm}^2$). One can see that a faithful reconstruction is produced with a power density of $23 \mu\text{W}/\text{cm}^2$.

Figure R4. Dynamic range of the spectrometer. **(a)** Photocurrent of the device as a function of 635 nm laser power density at $V_g = 20 \text{ V}$. **(b)** Transfer curves of the device in dark conditions (black) and under 635 nm laser illumination ($P_{in} = 23 \mu\text{W}/\text{cm}^2$, red). **(c)** The reconstructed 635 nm laser spectrum and the reference spectrum measured using a commercial spectrometer.

This important piece has been included in the supplementary information of the revision and discussed in main text as well. See Supplementary Fig. 11 in page 11 in supplementary information and from line 17 to line 19 in page 9. **“To get the dynamic range or the lowest irradiance, we measured the I - V_g (Supplementary Fig. 11b) and spectrum with lowest**

power density inside the group ($23 \mu\text{W}/\text{cm}^2$, Supplementary Fig. 11c). A faithful reconstruction is produced with a power density of $23 \mu\text{W}/\text{cm}^2$.”

Comment: It seems as far as I can tell that there is only one device tested here - what is the device to device repeatability? Even if not good, at the very least I would expect some reference to it. If it is satisfactory, while not essential, a demonstration of a rudimentary snapshot spectral imaging array (even just a 3×3 grid of devices for instance) would increase the importance of the work significantly.

Response: Thank you very much for raising this very valid concern. We measured around 10 devices. All of them show essentially similar capability to reconstruct visible light spectra to the device demonstrated in the main text. We also fabricated an imaging array as per your suggestion, using CVD grown SnS_2 and ReSe_2 (**Figure R5**). You can see that all devices work properly. However, some further work is required to optimize these CVD grown 2D materials for more uniform quality and better device performance. The objective is to continue to improve the performance of the imaging array.

Figure R5. (a, b, c) Optical microscopy images of the 3×3 $\text{SnS}_2/\text{ReSe}_2$ heterostructure array. Scale bar, (a) $200 \mu\text{m}$, (b) $20 \mu\text{m}$, (c) $50 \mu\text{m}$. (d) Transfer curves of devices shown in (c).

This important piece has been included in the supplementary information of the revision and discussed in main text as well. See Supplementary Fig. 12 in page 11 in supplementary information and from line 19 to line 23 in page 9. “**We measured around 10 devices, and all showed essentially similar capabilities to reconstruct visible light spectra. An imaging array was also fabricated, using chemical vapor deposition (CVD) grown SnS_2 and ReSe_2**

(Supplementary Fig. 12). Although all the devices worked properly, further optimization is needed to improve their performance.”

Comment: While the initial response time is on the order of 10s of milliseconds (at least, in “fast” mode), the PC still increases beyond this (and in “memory” mode, decreases) over the order of seconds. It strikes me therefore that it may be necessary to wait until the photoresponse has plateaued before the data can be collected? And then also this must be repeated at each value of gate voltage? If this is the case I imagine that the data collection time becomes significant – possibly on the order of 10s of seconds or more. Either way, for transparency, the rates of acquisition should at the very least be mentioned in the manuscript, especially if it is notably slow. To me, the current writing gives the impression that one measurement can be taken in a snapshot, rather than needing to scan across the different gate voltages.

Response: Thank you for raising this point, and you are indeed largely correct in your understanding. Practically, in this work, we continuously illuminated the device with light for 1 min to stabilize the photocurrent, and then swept the gate voltage from -20 V to 20 V (sweep rate 1 V/s) while recording the corresponding current values. Thus, the time required to measure a single spectrum is approximately 100 s. However, this duration can be reduced by decreasing the illuminating time, increasing the sweep rate, or/and using compressive sensing techniques. We have added a description of this operation process in the revision to clarify this point. See from line 5 to line 9 in page 9. **“Note, while recording the corresponding current values, we continuously illuminated the device with light for 1 min to stabilize the photocurrent, and then swept the gate voltage from -20 V to 20 V (sweep rate 1 V/s). Thus, the time required to measure a single complete spectrum is approximately 100 s, which can be reduced by decreasing the illuminating time, increasing the sweep rate, or/and using compressive sensing techniques.”**

Comment: If there is a persistent photocurrent after each illumination, does that mean that the gate voltage needs to be pulsed back and forth between each value to “reset” the device? Or can it simply be scanned across the range of values? If the former this should be mentioned.

Response: Thank you for highlighting this issue. As noted in the previous point, one spectrum can be reconstructed through a simple scanning of gate voltage from -20 V to 20 V to with no need to apply a gate pulse.

Comment: The figures (3a-e) and discussion concerning the compressive sensing is poorly explained even for me, and would be extremely unclear for anyone who is not familiar with compressive sensing. In general I would advise completely re-writing this section to make it more accessible, on top of some more specific points below:

Response: We would like to express our gratitude to the reviewer, for highlighting the parts of the manuscript needing greater care to improve the clarity. In response to this feedback, we have revised the compressed sensing section to help ensure its accessibility to readers of all backgrounds. We hope that these changes adequately address your concerns. **See from line**

8 to line 18 in page 11 and also from line 7 in page 20 to line 16 in page 21.

Comment: a. It is not clear to me (nor earlier in discussion of figure 2) in what sense you are referring to the Nyquist-Shannon theorem, given that we are not “sampling” the spectrum in a traditional manner. Further clarification in the text is necessary.

Response: Effectively, the Shannon-Nyquist theorem is primarily concerned with the sampling and reconstruction of time-dependent signals, specifically continuous-time signals or band limited discrete-time signals. It sets parameters for determining the minimum sampling rate to accurately reconstruct such signals. However, if we consider non-time-dependents signals as spatially dependent signals, such as images or spatial data, similar principles can be applied. In these cases, the sampling theorem may be extended to spatial sampling, where the sampling rate is determined based on the spatial frequency content of the signal. In our case, the sampling theorem is applied to the wavelength sampling. We have revised this section to enhance clarity for the readers. See from line 11 to line 14 in page 9. “ **Note, although the Shannon-Nyquist theorem is primarily concerned with the sampling and reconstruction of time-dependent signals, in this work, this theorem is extended to the spatial frequency content of the signal and applied to the wavelength sampling.**”

Comment: b. Fig 3a: the one sentence explanation of this figure in the text is unsatisfactory – it is not really clear why you have chosen 9 sensors, given that this doesn’t actually seem to be at the inflection point in the plot? In what sense is this “capturing 99% of the energy”? The non-specialised reader will also not know what a “singular value” is.

Response:

We would to extend our gratitude to the reviewer for the valuable feedback regarding the clarity of the text (compressed sensing). In response to the concerns, we have completely rewritten the section to ensure it is more accessible for non-specialized readers. Special attention has been given to defining all the parameters used within the context and with references, as the previous version lacked sufficient explanations and had the potential to confuse readers.

In this work, we utilized singular value decomposition (SVD), a well-known compression technique that effectively achieves a highly transformative basis while maximizing data sparsity, to determine the customized basis Ψ_r . In our case, a singular value (σ_i) represents the contribution of each feature of $R_{G,\lambda}$. We employed an optimal hard thresholding to determine the truncation rank (r) of the tailored basis, which allows us to capture the energy (in the submitted version) or the main features of $R_{G,\lambda}$ (in the revised version) while mitigating residual noise in the data. In Figure. 3a, we presented the singular value (σ_i) which represents the contribution of each feature and we estimated the information captured by the low-rank matrix

Ψ_r by the following equation $(\frac{\sum_{i=1}^r \sigma_i}{\sum_{i=1}^n \sigma_i} \times 100)$. To determine the optimal number of sensors p or gate voltage values, we followed methodology similar to that used by Manohar et al. with QR decomposition and column pivoting. The determination of the minimum sensor p was done by decreasing gradually the p value (from 81 to 9), and determining the mean square error (

), where represent the reconstructed photocurrent from the p sensors, the original photocurrent curves, and $n = 81$.

We have revised this section to enhance clarity for the readers. See from line 8 to line 18 in page 9. **“For this, we utilized the singular value decomposition (SVD) technique, a well-known compression technique to achieve full spectrum reconstruction from surprisingly few p measurements of I_G as described in the Methods section. In our case, a singular value (σ_i) represents the contribution of each feature of $R_{G,\lambda}$. To determine the optimal number of sensors p or, alternatively, distinct gate voltage values for measurements, we followed methodology similar to that used by Manohar et al. through QR decomposition and column pivoting. By gradually decreasing the p value from 81 to 9 and comparing the mean square error ($MSE = \frac{1}{n} \sum_{i=1}^n (I_G - \tilde{I}_G)^2$), where \tilde{I}_G represent the reconstructed photocurrent from the p sensors, I_G is the original photocurrent curve, and $n = 81$. Fig. 3a shows the singular values σ_r and the optimal modal is noted to occur for this system at $r = 9$, capturing 99% of the main features of $R_{G,\lambda}$ ($\frac{\sum_{i=1}^r \sigma_i}{\sum_{i=1}^n \sigma_i} \times 100$).”**

Comment: c. Figs 3b,c: It is extremely unclear what these measurement matrices actually represent and what conclusions should be drawn from them. What does the colour scale signify?

Response: In compressed sensing, the choice of measurement matrices is a critical to achieve accurate signal reconstruction. The matrices represent the selected Vg values to measure the photocurrent to enable reconstruction of a signal. Typically, the selection of the entries for measurement matrices are Gaussian or Bernoulli distributed random variables. Here, by using the work of Manohar et al., we find the optimized measurement matrices with a minimum number of sensors, thus decreasing the required number of measurements and increasing the speed of acquisition by a factor of nine. The colour scale in Figures 3b and 3c means 0 (not selected) or 1 (selected). We have replotted Figures 3b and 3c to make them clearer. See new Figure 3 in page 10.

Comment: Other more minor notes:

a. Fig. 1g: Captioned wrongly.

Response: Thank you very much for pointing out this mistake. We have corrected them in revision. See new caption of Fig. 1g in page 4.

Comment: b. Fig. 2f: It is unclear in what way you are “reconstructing” the photocurrent values here or more importantly, why, given that typically in this field (and within this paper mostly) reconstruction refers to measurement of light spectra.

Response: In Fig. 2f, the “reconstructed photocurrent” refers to the values obtained by the product of the spectrum vector (P_λ) measured using the commercial spectrometer and the

response matrix ($R_{G,\lambda}$) of the SnS₂/ReSe₂ device, i.e., $P_\lambda \times R_{G,\lambda}$. We are sorry for not defining the concept of “reconstruction” more rigorously, and we have deleted this figure to avoid the possibility of misleading readers.

Comment: c. Fig. 3b: No horizontal axis.

Response: We have added horizontal axis in revision. See new Fig. 3b in page 10.

Comment: d. Fig. 3h: All the works on this plot need to be referenced (i.e. the reference number labelled alongside the description of the class of device in the plot).

Response: According to your suggestion, all works in Fig. 3h have been referenced in the caption of Fig. 3h. See caption of Fig. 3h in page 10.

Comment: e. Fig. 5c: What does the dotted vertical line represent?

Response: In Fig. 5c, the dotted vertical lines represent the times when the laser pulse and gate pulse are applied, and we have added corresponding description in the caption. See caption of Fig. 5c in page 15.

Comment: References

[1] Deng, W. et al. Electrically tunable two-dimensional heterojunctions for miniaturized near-infrared spectrometers. Nat Commun 13, 4627 (2022).

[2] H. H. Yoon et al. Miniaturized spectrometers with a tunable van der Waals junction. Science 378, 296–299 (2022).

[3] Xiang, D. et al. Two-dimensional multibit optoelectronic memory with broadband spectrum distinction. Nat Commun 9, 2966 (2018).

Response: Thank you for your constructive suggestions. These valuable piece of literature have been cited and discussed in revision. See Reference 11 and 15 in page 23 and Reference 45 in page 25.

Reviewer #3 (Remarks to the Author):

Comment: The article titled "Miniaturized spectrometer with intrinsic long-term image memory" by Wu G. et al. introduces a miniaturized spectrometer that utilizes a 2D SnS₂/ReSe₂ van der Waals heterostructure. The device is capable of photodetection, spectrum reconstruction, and spectral imaging within the 400-800 nm bandwidth with a spectral resolution of 5 nm. Additionally, the non-volatile storage of photo-generated carriers with long retention characteristics of 10⁴ s is observed in the heterostructure spectrometer due to interfacial trap

states. This work presents a straightforward method to achieve a single non-volatile spectrometer by stacking atomic layered materials, which could be of interest to the 2D optoelectronics and image processing communities in Nature Communications. However, several key conclusions in this work lack sufficient support from experimental observations and simulation results. The mechanism behind the non-volatile heterostructure spectrometer's operation remains unclear, and its potential application beyond von Neumann architectures is uncertain. I recommend that the authors address these issues before publishing this work.

Previously, single spectrometers based on 1D and 2D materials have been reported, exhibiting both wide bandwidth and high spectral resolution (Science 365, 1017–1020 (2019); Nano Lett., 20, 320–328 (2020); Science 378, 296–299 (2022); Nat. Commun. 13:4627 (2022)). In an attempt to differentiate their work from these reports, the authors assert that their SnS₂/ReSe₂ spectrometer, operating in a non-volatile manner, could be applied beyond von Neumann architectures. However, the simple demonstration of the imaging memory system in Fig. 5 does not adequately reveal its potential use in non-von Neumann architectures such as artificial neural networks or machine vision. To strengthen the motivation of their work and differentiate their spectrometer from previous reports, the authors should provide additional experimental results and discussions on how their heterostructure spectrometer can be utilized in non-von Neumann architectures.

Response: Thank you very much for highlight the weak aspects of the manuscript, and we now recognize that we had not provided enough evidence to show how their heterostructure spectrometer could be utilized in non-von Neumann architectures. According to your suggestion, in this revised version, we have performed additional experiments to showcase that our miniaturized spectrometer not only functions as memory storage but also has neuromorphic computing capabilities (**Figure R1**). To demonstrate this, periodic light pulses ($P_{in} \sim 10 \text{ mW/cm}^2$, pulse-width $\sim 1 \text{ s}$) were applied to the device to produce multiple memory states for data storage, and periodic gate pulses (amplitude $\sim 5 \text{ V}$, pulse-width $\sim 2 \text{ s}$) were subsequently applied to erase the stored data. The observed long-term potentiation and depression cycle, suggests that our device mimics a synapse with a nonlinear behavior for training the Artificial Neural Network. Indeed, we obtained a classification accuracy up to 90% for 50 epochs, which opens up possibilities for the development of a multifunctional sensor (camera), which can be used as storage and as neuromorphic computing. In fact, our attempt also suggests that, by utilizing Deep Neural networks, we can potentially reconstruct the spectrum of the incident light with a certain number of memory cells since the long-term potentiation shape shows a dependency on the incident light. Thus, our proposed miniaturized spectrometer can go beyond von Neumann architecture.

These important discussion points have added to revision as well. See Figure 6 in page 17 and from line 11 in page 17 to line 10 in page 18. **“Optical memory effects have been reported by Xiang et al.. We would like to stress that our device's memory function is due to**

interface trap states, making it promising for applications in artificial synapses, neural networks, machine vision and imaging processing. To showcase that our miniaturized spectrometer not only serves as storage but also has the processing capabilities. Periodic light pulses ($P_{in} \sim 10 \text{ mW/cm}^2$, pulse-width $\sim 1 \text{ s}$) were applied to the device to produce multiple memory states for data storage, and periodic gate pulses (amplitude $\sim 5 \text{ V}$, pulse-width $\sim 2 \text{ s}$) were then applied to erase the stored data (Fig. 6a). Long-term potentiation and depression cycles were observed, suggesting that our device mimics the function of a synapse with a nonlinear behavior for training the Artificial Neural Network (ANN) (Fig. 6b). Moreover, the long-term potentiation shape shows a dependency on incident light, suggesting that the possibility to reconstruct the spectrum of incident light with a defined number of memory cells (Fig. 6c). We simulated a two-layer ANN based on the $\text{SnS}_2/\text{ReSe}_2$ vdW heterostructures synaptic behavior to classify handwritten data on the MNIST data (Fig. 6d). Details of the simulation can be found in the Method section. Fig. 6e and 6f show the classification accuracies for different states (from 10 to 40 states). As the number of states increases from 10 to 40, the classification accuracy slightly increases from 87 % until it reaches nearly 90% after only 50 epochs for 40 states. Moreover, the greater the number of the states, the shorter the training time. ”

Comment: The $\text{SnS}_2/\text{ReSe}_2$ device exhibits long-term positive persistent photoconductivity, which is distinct from the fast photoresponse behavior reported in the $\text{MoS}_2/\text{WSe}_2$ heterostructure (Science 378, 296–299 (2022)). The authors claim that such non-volatile photoresponse is induced by the interfacial trap states in the $\text{SnS}_2/\text{ReSe}_2$ heterostructure. However, it remains unclear how these trap states are introduced in the heterostructure compared to the $\text{MoS}_2/\text{WSe}_2$ device. Additionally, it is essential to understand how the trap states affect the dynamic transport behavior of photo-generated carriers in the heterostructure, which is likely more complex than the explanation provided for a single material in Fig. S8. Furthermore, it would be helpful to know if there are any controllable methods for introducing trap states in the $\text{SnS}_2/\text{ReSe}_2$ heterostructure, which could consistently induce such non-volatile photoconductivity. To validate their claim of interfacial trap states induced long-term photoconductivity, the authors should provide more explanations and experimental results.

Response: Thank you for highlighting the lack of explanation for the interface trap states, and we agree with you that clarifying and finding a controlled method to introduce these trap states are important. According to your suggestion, we performed EDX characterization of ReSe_2 sample and found the presence of Re vacancies with an atomic ratio of $\sim 2.5 : 1$ between Se and Re (Figure R6a). In our $\text{SnS}_2/\text{ReSe}_2$ heterostructure, the trap states are thought to be mainly introduced by the Re Vacancies in ReSe_2 . Figure R6b schematically shows the band profile between overlapping and non-overlapping regions of the structure, where trap states (orange solid line) in the energy gap are formed by Re vacancies. Under laser illumination, electron-hole pairs are generated and the electrons in ReSe_2 are subsequently trapped by the

trap states induced by Re vacancies, resulting in a photogating effect. Since the electron-hole pair generation depends on the wavelength (absorption), photoexcited carrier trapping at the interface also depends on the wavelength. For photodetectors dominated by the photo-gating effect, the photocurrent (I_{ph}) can be written as $I_{\text{ph}} = \frac{\partial I_{\text{ds}}}{\partial V_{\text{g}}} \Delta V_{\text{g}}$, where ΔV_{g} is the local gate voltage generated by the photoexcited carrier trapping at the interface³⁰. **Fig. 1m** plots the calculated ΔV_{g} as a function of the light wavelength for a variety of gate voltages. Interestingly, ΔV_{g} has the same trend as the photocurrent (**Fig. 1f**), indicating a strong dependence of the photogating effect on the light wavelength and gate voltage. This is also key to the electrically tunable memory effect discussed further in the paper. This photocurrent can be maintained even when the light is turned off. This state can be eliminated by raising the Fermi level with an applied gate pulse (red dashed line in **Figure R6b**), under which trapped charges will be released.

Our recent work suggests that trap states can be introduced by the gas adsorption. Since ReSe₂/SnS₂ and InSe/SnS₂ have similar band profiles (the experimental details of InSe/SnS₂ can be found in the literature of ACS Nano 16, 17347–17355 (2022)), the structure of ReSe₂/SnS₂ is similar to that of InSe/SnS₂ have the same memory behavior. We would like to stress that the wavelength-dependent absorption of the ReSe₂ and SnS₂ materials and the gated carrier transport through the interface are similar to what occurs in a MoS₂/WSe₂ heterojunction. Thus, gas adsorption provides additional band-profile modulation and memory effects. These effects should also be present in other vdW heterojunctions.

Figure R6. (a) EDX characterization indicating the presence of Re vacancy. (b) Mechanistic understanding of the optoelectronic memory operation.

These important discussion points have added to revision as well. See **Supplementary Fig. 6** in page 6 in Supplementary Information and from line 4 to line 9 in page 6 in main text. “**Energy-dispersive X-ray spectroscopy (EDX) characterization of ReSe₂ sample suggests the presence of Re vacancies with an atomic ratio of ~2.5:1 between Se and Re (Supplementary Fig. 6a). Fig. 1l schematically shows the band profile between overlapping and non-overlapping regions of the structure, where trap states (orange solid line) in the energy gap are formed by Re vacancies. Under laser illumination,**

electron–hole pairs are generated and the electrons in ReSe₂ are subsequently trapped by the trap states induced by Re vacancies, resulting in a photogating effect. Since the electron–hole pair generation depends on the wavelength (absorption), photoexcited carrier trapping at the interface also depends on the wavelength.”

Comment: Although the trap states induced photogating effect dominates the SnS₂/ReSe₂ heterostructure photodetector, it is still unclear why the device exhibits a peak when the wavelength is changed in the visible range. To clarify the working mechanism of the heterostructure spectrometer, the authors should provide more experimental results and discussions. Additionally, besides the SnS₂/ReSe₂ and MoS₂/WSe₂ heterostructures, it would be interesting to investigate whether similar spectrometers can be achieved by stacking other 2D materials. Furthermore, it is essential to explore whether there are any specific requirements regarding the band offsets in the heterostructures.

Response: Thank you for your question about peak wavelengths in the visible range. **Figure R7a** shows how three photovoltaic voltages are generated in the structure during illumination. The voltages of V_{pv1} and V_{pv3} are connected forward to the external bias in the circuit, while V_{pv3} is connected in reverse. When a light spot is focused on the ReSe₂ or SnS₂ regions, the total current increases monotonically due to the monotonic increase of the photovoltaic voltage of V_{pv1} or V_{pv3} , as shown in Fig. 1i and Fig. 1k in the manuscript. In the overlapping region, the efficiency at which the photovoltaic voltage V_{pv2} is generated depends on the wavelength of the light. The efficiency is high when the photon energy is larger than the energy gap of SnS₂, and the efficiency is low when the photon energy is smaller than the energy gap. Given that the reverse photovoltaic voltage offsets a portion of the external bias and reduces the total current, the peak photocurrent occurs around a wavelength of ~580 nm (gap energy of SnS₂) as shown in Fig. 1j in the manuscript. The conduction-band offset at the heterojunction has the effect of resisting the photoelectrons transferred from SnS₂ to ReSe₂, further decreasing the efficiency of V_{pv2} generation and shifting the wavelength peak to lower optical wavelengths. Conversely, a smaller conduction-band offset results in longer wavelength peaks. Therefore, the absorption peak is tunable because the conduction band offset is tunable, which can be achieved by changing the gate voltage, as shown in Figure 1f in the manuscript. We also calculated photocurrent as a function of light wavelength at various gate voltages considering the competition among the three photovoltaic voltages around the interface. Similar effect has been observed (**Figure R7b**). Although both MoS₂/WSe₂ and ReSe₂/SnS₂ heterojunctions have gate-tunable effects, the large-band offset in ReSe₂/SnS₂ leads to monotonic behavior of the photoconductive I_{ds} - V_g characteristics, which is inconsistent with the tunable ambipolar behavior for MoS₂/WSe₂ (low-band offset heterojunction).

Figure R7. (a) Schematic of the mechanism of peak wavelengths in the visible range. (b) Calculated photocurrent as a function of light wavelength at various gate voltages from -5 V to 60 V.

These important discussion points have added to revision as well. See **Supplementary Fig. 7** in page 7 in Supplementary Information and from line 9 to line 15 in page 6 in main text. **“Moreover, three photovoltaic voltages are generated in the structure during illumination and the competition among these three photovoltaic voltages around the interface can cause the variation of the photocurrent peak wavelength (Supplementary Fig. 7).”**

Comment: It would be beneficial to demonstrate laser spectra at other wavelengths ranging from 400 nm to 800 nm in addition to the 635 nm laser spectrum reconstructed and measured using a commercial spectrometer in Fig. 2g. The same applies to the measurements in Fig. 3f, where it would be useful to illustrate more spectra with different wavelengths.

Response: Thank you very much for this good suggestion. We have reconstructed a series of quasi-monochromatic spectra in the range of 400~800 nm with our single-heterostructure spectrometer, which agree well with the reference spectra using a commercial spectrometer (**Fig. R3**). And we have also demonstrated the reconstruction of 632 nm and 532 nm laser spectra in **Fig. 3f** and **Fig. R8** respectively.

Figure R8. Reconstructed 532 nm laser spectrum using optimized compressed sensing method and the reference spectrum measured using a commercial spectrometer.

These important discussion points have added to revision as well. See **Supplementary Fig. 10** in page 10 in Supplementary Information and from 3 to line 4 in page 9 in main text. “ **We also demonstrated the reconstruction of 532 nm and 635 nm laser spectra in Supplementary Fig. 10 and Fig. 2g respectively.**”

REVIEWER COMMENTS

Reviewer #1 (Remarks to the Author):

My comments have been properly addressed in the revision, and the manuscript has been improved quite noticeably. I recommend this work for publication.

Reviewer #2 (Remarks to the Author):

Thank you to the authors for responding to my comments in a thorough manner. Many of them have been addressed satisfactorily, namely:

- The demonstration of full operational wavelength range.
- The attempt at making an image sensor, which showcases the repeatability of the process for multiple devices, even if not a working image sensor.
- The clarification added regarding the length of time for one “full” measurement to reconstruct an unknown spectrum, as well as the lack of need for gate pulses between each measurement.
- Explanation around their interpretation / use of the Nyquist-shannon theorem.
- The discussion around compressive sensing is vastly improved, for which I commend the authors; I believe this section is now far more accessible to the more general reader.
- All the “minor notes” have been corrected as advised.

I still have some outstanding concerns regarding the original issues I raised, that need to be addressed:

1. The extra detail on the power response is a good addition that improves the transparency of the work. However, the wording is somewhat confusing now – what has been measured is the lowest irradiance at which a spectrum can be reconstructed, not the dynamic range (it currently says the dynamic range or lowest irradiance) – this should be reworded to make clear. It is also not clear what the authors mean by “inside the group” – I presume this simply means, within the apparatus available to them; if so, fine, but please edit accordingly.

2. I believe now the additional demonstrations potentially differentiate this work from it sufficiently, but it is worth noting the device itself is still, in design, operation and function, essentially the same as that in H. H. Yoon et al Science 378, 296–299 (2022). From their response to my comment, the authors seem to want to differentiate it by emphasising the importance of trap states, but as far as I can tell, this is only responsible for the optical memory effect – the actual gate tunability itself is mostly attributable to photovoltaic effects at the junction (as is the same with Yoon et al). This is something they indeed themselves discuss in supplementary figures 6 and 7, respectively. I would invite the authors to make sure that in the main text there is clarity as to which mechanism they believe to be involved / mainly responsible for which functionality of the device, especially around lines 110-120.

3. My main concerns are around figure 6. I believe that if explained properly, alongside the demonstration of memory-recalled spectral reconstruction shown in figure 5, the work as a whole represents a sufficient novelty on past literature. However, while I think the data / results provided are probably suitable, at the moment the explanation regarding how the device actually functions with respect to an ANN is far from adequate, to the extent that I cannot assess its significance.

In particular:

- The practical measurement process by which device data could be used to train/execute such an ANN is not clear. I believe I am right in saying that from fig 6d onwards we are just dealing with simulated data, but it is not clear what data from the device is even being used here. I presume that the spectrometer would be scanned across the image such that each pixel is essentially “measured” by the spectrometer, and the current values then feed into the ANN, but this is not clearly written, nor does it convey how the synaptic weight is adjusted – authors should please make a strong effort to improve the clarity here, and potentially to move some of the information from the methods to the main text.

- Many device physicists who would be interested in the paper will not necessarily be familiar with machine learning and as such the authors should try to make the terminology as accessible as possible. The authors should be careful for instance that terms / concepts (e.g. epoch, synaptic weight, confusion matrix) are actually explained when they are first introduced, so that readers can easily understand the impact of the demonstration rather than having to look up terms themselves.

- On a more minor note, in 6f it was not immediately clear to me that the values here being referred to are the handwritten digits – I would change the axis labels to something like predicted / actual handwritten digit.

Reviewer #3 (Remarks to the Author):

The authors have conducted supplementary experiments and simulations to support their conclusions regarding the device’s working mechanism and its neuromorphic applications. However, the two-layer ANN simulation, relying on P-D curves for MNIST dataset recognition, lacks novelty as it has already been extensively reported. Moreover, the operational mechanism of the non-volatile heterostructure spectrometer remains unclear, and certain explanations in the rebuttal letter is inconsistent with those presented in the initial manuscript. Therefore, I cannot recommend the publication of this paper in Nature Communications unless the authors elucidate the device's working mechanism and resolve the raised concerns outlined below.

1. In the rebuttal letter, the authors claim that the presence of Re vacancies induces trap states in ReSe₂ flake, which leads to the photogating and non-volatile memory effect. However, this clarification is inconsistent with the content in the manuscript, where it is stated that both the trap states in ReSe₂ and SnS₂ flakes could contribute to the observed effect. According to the manuscript, the photo-generated

holes and electrons are trapped in SnS₂ and ReSe₂ respectively, leading to the photogating effect. To validate the defects induced photogating effect in the heterostructure, it is crucial to conduct the first-principle calculation, which could elucidate the distribution of trap states both in SnS₂ and ReSe₂ and their impact on photo-carrier transport and storage in the heterostructure. Moreover, the characterizations including TEM and XPS should be conducted to confirm the types and concentration of defects in SnS₂ and ReSe₂, which are more precise and reliable than the EDX in the rebuttal letter. Additionally, apart from the defects present in the individual flakes, the interfacial trap states are generally observed in 2D heterostructures during stacking process, which might also trap photo-generated carriers and prolong the carrier lifetime. The authors should also clarify whether such interfacial traps contribute to the photogating effect in their devices.

2. To elucidate the peak current in the SnS₂/ReSe₂ spectrometer, the authors present a band alignment diagram involving three photovoltaic voltages generated in the ReS₂, overlapped SnS₂/ReSe₂, and SnS₂ regions under light illumination. However, it is still not clear how these photovoltaic voltages compete with each other as the wavelength changes, leading to the emergence of peak current. Also, why is the peak position only determined by the bandgap of SnS₂, as claimed by the authors in the rebuttal letter? To address these uncertainties, the authors should provide a more quantitative analysis based on the experimental results and the proposed model.

The inclusion of additional simulations based on a two-layer ANN lacks novelty in demonstrating the spectrometer's potential in non-von Neumann architecture. I think that further investigation into the mechanism of the spectrometer is necessary to enhance the insights gained from this work.

Response to reviewers' comments

Reviewer #1 (Remarks to the Author):

My comments have been properly addressed in the revision, and the manuscript has been improved quite noticeably. I recommend this work for publication.

Response: We greatly appreciate your affirmation of this work and for raising many useful and valuable suggestions.

Reviewer #2 (Remarks to the Author):

Comment: Thank you to the authors for responding to my comments in a thorough manner. Many of them have been addressed satisfactorily, namely:

- The demonstration of full operational wavelength range.
- The attempt at making an image sensor, which showcases the repeatability of the process for multiple devices, even if not a working image sensor.
- The clarification added regarding the length of time for one “full” measurement to reconstruct an unknown spectrum, as well as the lack of need for gate pulses between each measurement.
- Explanation around their interpretation / use of the Nyquist-shannon theorem.
- The discussion around compressive sensing is vastly improved, for which I commend the authors; I believe this section is now far more accessible to the more general reader.
- All the “minor notes” have been corrected as advised.

I still have some outstanding concerns regarding the original issues I raised, that need to be addressed:

Response: We greatly appreciate your affirmation of this work, and for expressing many useful and valuable opinions. Responses to your outstanding questions and concerns are listed as follows in a point-by-point manner.

Comment: The extra detail on the power response is a good addition that improves the transparency of the work. However, the wording is somewhat confusing now – what has been measured is the lowest irradiance at which a spectrum can be reconstructed, not the dynamic range (it currently says the dynamic range or lowest irradiance) – this should be reworded to make clear. It is also not clear what the authors mean by “inside the group” – I presume this simply means, within the apparatus available to them; if so, fine, but please edit accordingly.

Response: Thank you very much for pointing out how this could possibly be misleading to readers. According to your suggestion, we have clarified this point in the revision. See from line 5 to line 7 in page 10. **“We used the lowest irradiance suppliable by the apparatus**

available to us (23 $\mu\text{W}/\text{cm}^2$) and a faithful reconstruction was produced with such low power density.”

Comment: I believe now the additional demonstrations potentially differentiate this work from it sufficiently, but it is worth noting the device itself is still, in design, operation and function, essentially the same as that in H. H. Yoon et al Science 378, 296–299 (2022). From their response to my comment, the authors seem to want to differentiate it by emphasizing the importance of trap states, but as far as I can tell, this is only responsible for the optical memory effect – the actual gate tunability itself is mostly attributable to photovoltaic effects at the junction (as is the same with Yoon et al). This is something they indeed themselves discuss in supplementary figures 6 and 7, respectively. I would invite the authors to make sure that in the main text there is clarity as to which mechanism they believe to be involved / mainly responsible for which functionality of the device, especially around lines 110-120.

Response: Thank you for bringing this to our attention. We concur with your observation that distinguishing between photovoltaic effects and photogating effects can be challenging. Photovoltaic effects arise from carriers accumulating at the interface, while a photogating effect is a result of carriers trapped at the interface. In order to distinguish between photovoltaic effects and photogating effect, we conducted TEM and XPS characterization to determine the defects' nature and concentration, along with first-principle calculations. Based on TEM and XPS analysis and characterization, it was determined that our SnS_2 possesses ~2% sulfur vacancies, while the ReSe_2 has ~2.4 % Re vacancies. First-principle calculations revealed that removing one Re atom resulted in the appearance of three defect bands labeled DF_{R1} (0.26 eV below bottom of conduction band (CB)), DR_{R2} (0.49 eV below the bottom of CB), and DF_{R3} (0.37 eV above the top of the valence band (VB)). In the case of SnS_2 with a sulfur vacancy, a single defect band (DF_{S}) emerged, positioned 0.7 eV below the bottom of the CB. Moreover, due to the upward bending of the energy band, hole trapping plays a unique role in the overlapping region as excited holes will move to the overlapped region and electrons will move to the nonoverlapped region. Thus, under illumination, holes are excited and move to the overlapping region and some of the holes will be trapped by DF_{R3} , resulting a photogating effect, which would have the effect of enhancing the photocurrent. When the light is switched off, the trapped holes remain and sustain the photocurrent, resulting in a memory effect as you mentioned. We would like to stress that the number of holes trapped depends on V_{biR} . Increasing V_{g} , V_{biR} decreases, resulting in a decrease in number of trapped holes, which is consistent with the experimental results. For photodetectors dominated by a photo-gating effect, the photocurrent in the interface ($I_{\text{interface}}$) can be written as $I_{\text{interface}} = \frac{\partial I_{\text{ds}}}{\partial V_{\text{g}}} \Delta V_{\text{g}}$, where ΔV_{g} is the local gate voltage generated by photoexcited carrier trapping at the interface. In Figure 1m of main text, the calculated ΔV_{g} is shown to decrease with increasing gate voltage.

To understand the peak observed in the photocurrent, we have to consider the photogating effect and the photo current generated in the ReS₂ (I_R) and the SnS₂ regions (I_S). The total photocurrent (I_{ph}) has three main contributions: the photocurrents generated in the ReS₂ (I_R) region, SnS₂ region (I_S), and the photocurrent generated in the overlapping region ($I_{interface}$) due to the photogating effect. As the Au electrode has a much greater work function compared with that of SnS₂ and ReSe₂, ϕ_{B1} is greater than V_{biR} . Thus, I_S and I_R are in opposite directions, and thus have opposing contributions. Therefore, $I_{ph} = I_S - I_R + I_{interface}$. Moreover,

$I_{interface} = \frac{\partial I_{ds}}{\partial V_g} \Delta V_g$ is almost constant when far from the photocurrent peak (15 V in **Figure**

R6c). Thus, the peak of photocurrent can be explained by the $|I_S - I_R|$. Figure R6b plots the R-S and it does show a peak around 600 nm, which is consistent with the photocurrent of the device. We can further write the photocurrents generated in the ReS₂ and SnS₂ regions as: $I_i = qG_i(\lambda)\tau_i\mu_iSV_{bii}/W_{Di}$, where I_i , $G_i(\lambda)$, τ_i , μ_i , V_{bii} , and W_{Di} are the photocurrent, photogeneration rate, lifetime, mobility, effective built-in potential, and effective depletion width in region i , respectively. Moreover, the built-in potential of the SnS₂ side decreases the most with increasing V_g and that of the ReSe₂ side remains quite stable (Figure R4e). In other words, photocurrent decreases much faster with increasing V_g . Thus, the peak position of the photocurrent moves to longer wavelength with increasing V_g , which is consistent with the experimental observations (Figure R6c).

Thus, the photogating effect due to defects enhances the photocurrent and results in a memory effect and the peak of photocurrent can be qualitatively explained by $|I_S - I_R|$.

According to your suggestion, we have clarified this point in the revision. See from line 6 in page 6 to line 14 in page 7 in main text and also Supplementary Note 1 and Supplementary Note 2 in supporting information. **“TEM analysis and XPS characterization suggest that our SnS₂ possesses ~2% sulfur vacancies, while the ReSe₂ has ~2.4 % Re vacancies (Figs. 1c and 1d and Supplementary Fig. 6). First-principle calculations reveal Re vacancies result in the appearance of three defect bands labeled DF_{R1} (0.26 eV below bottom of conduction band (CB)), DR_{R2} (0.49 eV below the bottom of CB), and DF_{R3} (0.37 eV above the top of the valence band (VB)). In the case of SnS₂ with sulfur vacancies, a single defect band (DF_S) emerges, positioned 0.7 eV below the bottom of the CB (Supplementary Fig. 7). Moreover, due to the upward bending of the energy band, hole trapping plays a unique role in the overlapping region as excited holes will move to the overlapped region and electrons move to the nonoverlapped region (Fig. 1l and Supplementary Fig. 8). Thus, under illumination, holes are excited and move to the overlapping region and some of the holes will be trapped by DF_{R3}, resulting a photogating effect, which would have the effect of enhancing the photocurrent. When**

the light is switched off, the trapped holes remain and sustain the photocurrent, resulting in a memory effect. This is also key to the electrically tunable memory effect, discussed further in the paper. Moreover, the interfacial trap states, such as neutral traps (NT), are generally observed in 2D heterostructures during the stacking process, which also result in photogating and memory effects (Supplementary Fig. 9). Details of the calculation can be found in Supplementary Note 1. For photodetectors dominated by a photogating effect. the photocurrent in the interface ($I_{\text{interface}}$) can be written as

$$I_{\text{interface}} = \frac{\partial I_{\text{ds}}}{\partial V_g} \Delta V_g, \text{ where } \Delta V_g \text{ is the local gate voltage generated by photoexcited carrier}$$

trapping at the interface. In Fig. 1m, the calculated ΔV_g is shown to decrease with increasing gate voltage. We would like to stress that the number of holes trapped depends on built-in potential at ReSe₂ (V_{biR}). Increasing V_g , V_{biR} decreases, resulting in a decrease in the number of trapped holes, which is consistent with the experimental results. The total photocurrent (I_{ph}) has three main contributions: the photocurrents generated in the ReS₂ (I_R) and SnS₂ regions (I_S), and the photocurrent generated in the overlapping region ($I_{\text{interface}}$) due to the photogating effect. As the Au electrode has a much greater work function in comparison with both SnS₂ and ReSe₂, the built-in potential between Au and ReSe₂ (ϕ_{B1}) is greater than V_{biR} . Thus, I_S and I_R are in opposite directions. Therefore, $I_{\text{ph}} = I_S - I_R + I_{\text{interface}}$. Moreover, $I_{\text{interface}} = \frac{\partial I_{\text{ds}}}{\partial V_g} \Delta V_g$ is

almost constant when far from the photocurrent peak (15 V in Fig. 1m). Thus, the peak of photocurrent can be explained by $|I_S - I_R|$. Fig. 1j plots the R-S and it does show a peak around 600 nm, which is consistent with the photocurrent of the device. Moreover, the built-in potential of the SnS₂ side decreases the most with increasing V_g and that in the ReSe₂ side remains quite stable (Supplementary Fig. 8e). In other words, the photocurrent decreases much faster with increasing V_g (Supplementary Note 2). Thus, the peak position of the photocurrent moves to longer wavelength with increasing V_g , which is consistent with the experimental observation (Fig. 1m). "

Comment: My main concerns are around figure 6. I believe that if explained properly, alongside the demonstration of memory-recalled spectral reconstruction shown in figure 5, the work as a whole represents a sufficient novelty on past literature. However, while I think the data / results provided are probably suitable, at the moment the explanation regarding how the device actually functions with respect to an ANN is far from adequate, to the extent that I cannot assess its significance.

In particular:

The practical measurement process by which device data could be used to train/execute such an ANN is not clear. I believe I am right in saying that from fig 6d onwards we are just dealing

with simulated data, but it is not clear what data from the device is even being used here. I presume that the spectrometer would be scanned across the image such that each pixel is essentially “measured” by the spectrometer, and the current values then feed into the ANN, but this is not clearly written, nor does it convey how the synaptic weight is adjusted – authors should please make a strong effort to improve the clarity here, and potentially to move some of the information from the methods to the main text.

Response: First and foremost, we extend our gratitude to the reviewer for the valuable comments. We acknowledge that our previous explanation of the weight adjustment occurring during the training of the artificial neural network (ANN) might not have been easily comprehensible for a broader audience. **In fact, we are using the measured data with different numbers of cycles for training.** In the revised version, we have enhanced our explanation, offering additional details and insights into the dual functionality of our devices, serving as both neurons and synapses for training a basic neural network. In the training of our ANN, we utilize the measured experimental memory states controlled by either laser or by gate voltage pulses, to act as synaptic weights within a simple ANN framework. Typically, attaining high classification accuracy involves the utilization of artificial synapses exhibiting linear characteristics. In contrast, our approach relies on non-linear Long-Term Potentiation (LTP) triggered by blue light stimuli and Long-Term Depression (LTD) triggered by a gate voltage stimuli. To achieve a high classification accuracy, it was essential to consider the inherent asymmetric non-linearity of our LTP and LTD, along with the entire plasticity range represented by the I_{max}/I_{min} ratio.

In order to achieve this, we used an asymmetric nonlinear relationship to fit the LTP (1) and LTD (2) curves, thereby deriving the nonlinearity (NL) and the weight change of LTP and LTD during the weight update process.

$$I_p(N) = I_{min} + B \times \left(1 - e^{-\left(\frac{N}{A_p}\right)} \right) \quad (1)$$

$$I_D(N) = I_{max} - A \times \left(1 - e^{-\left(\frac{N-N_{max}}{A_D}\right)} \right) \quad (2)$$

$$B = (I_{max} - I_{min}) \left(1 - e^{-\left(\frac{N_{max}}{A_{p,D}}\right)} \right) \quad (3)$$

where I_p and I_D are functions describing the current of the potentiation and depression curves, respectively. I_{max} , I_{min} and N represent the maximum current, minimum current, and number of applied pulses, respectively.

Figure R1. (a) LTP and LTD curves of the vdW SnS₂/ReSe₂ device for 10 cycles. The excitatory pulse is a blue light (pulse-width ~1 s) with $P_{in} \sim 10$ mW/cm², whereas the inhibitory V_g with an amplitude of +5 V. (b) A single LTP/LTD curve of vdW SnS₂/ReSe₂ synaptic device, where the asymmetric nonlinearity factor (ANL) is 0.35.

Furthermore, the non-linearity is quantitatively determined by using the following the relation (4):

$$ANL = \left[\frac{I_P(N/2) - I_D(N/2)}{I_{max} - I_{min}} \right]. \quad (4)$$

Potentiation and depression curves that are entirely symmetric result in an ANL (Asymmetry of Nonlinearities) value of zero, whereas our device exhibits an ANL value of 0.35. In order to showcase the neuromorphic computing capabilities based on our vdW SnS₂/ReSe₂ devices, which function as both memory and synapses, we employed a simple neural network for a multiclass classification problem. The neural network was composed of two layers: an input layer with a dimension of 784 (representing each pixel in the 28x28 images), and an output layer with 10 neurons, each dedicated to recognizing a specific label ($f_{m,n}$). We trained and tested our neural network by using the MNIST dataset, which contains handwritten digits (0, ..., 9).

The training dataset consisted of 60,000 images, while the testing dataset contained 10,000 images. During the training process, each input neuron in the input layer received the corresponding pixel value from the image, assigns it to input vector (X_i), and converts it into 10 outputs values using the linear relation represented as $\Sigma_n = \sum_{i=1}^{784} X_i W_{i,n}$, where $W_{i,n}$ denotes the synaptic weight matrix. Initially, the weight matrix was randomly initialized based on our vdW SnS₂/ReSe₂ synaptic states. To determine the direction of the weight updates, the output values Σ_n were transformed by using a Softmax activation function $\left(\sigma(\Sigma)_i = \frac{e^{\Sigma_i}}{\sum_{j=1}^K e^{\Sigma_j}} \right)$,

resulting in a probability distribution denoted as ($O_{m,n}$). When we compared the label value ($f_{m,n}$) of each image 'm' with the probability distribution, it allowed us to calculate the delta value $\delta_{m,n} = O_{m,n} - f_{m,n}$. If $\delta_{m,n}$ is greater than 0, the synaptic weight (w) is increased; otherwise, the synaptic weight is decreased. The magnitude of the weight changes (Δw) was determined by the above fitting formulas.

According to your suggestion, we have clarified this point in the revision. See from line 8 in page 18 to line 11 in page 19 in main text and also Supplementary Note 3 in supporting information. **"To demonstrate the multifunctionality of our compact spectrometer, which not only serves as a storage device but also possesses processing capabilities. We applied periodic light pulses ($P_{in} \sim 10$ mW/cm², pulse-width ~ 1 s) to produce an increment of the conductivity, leading to multiple memory states for data storage. Subsequently, by applying periodic gate voltage pulses (amplitude ~ 5 V, pulse-width ~ 2 s), we could decrement the conductivity, thus performing erasure the stored states (Fig. 6a). In this context, the increase and decrease in device conductance are termed as potentiation and depression, respectively. Given that our vdW SnS₂/ReSe₂ device system changes conductance, it can thus be utilized in a neural network and act as both a neuron and a synapse. Long-term potentiation (LTP) and depression (LTD) cycles were observed, suggesting that our device mimics the function of a synapse with a nonlinear behavior for training the Artificial Neural Network (ANN) (Fig. 6b and Supplementary Fig. 17). Generally, achieving high classification accuracy often relies on the use of artificial synapses that have linear characteristics. However, in our case, we rely on non-linear LTP induced by blue light stimuli and LTD by voltage gate pulses. To attain a high classification accuracy, it is essential to consider the inherent asymmetric non-linearity of our LTP and LTD as well as considering the complete plasticity range represented by the ratio I_{max}/I_{min} . To accomplish this, we used an asymmetric nonlinear relationship to fit the LTP and LTD curves, thereby deriving the nonlinearity (NL) and the weight change of the LTP and LTD, the details are further discussed in Supplementary Note 3. Moreover, the long-term potentiation shape shows a dependency on the incident light, suggesting the possibility of reconstructing the spectrum of incident light with a defined number of memory cells (Fig. 6c). In this study, we simulated a two-layer ANN based on the SnS₂/ReSe₂ vdW heterostructures synaptic behavior to classify handwritten data on the MNIST data (Fig. 6d). Details of the training process can be found in the Methods section. Figs. 6e and 6f present the classification accuracies as function of epoch (which represents a single iteration over the complete training dataset) and varying the numbers of synaptic states (ranging from 10 to 40 states). As the synaptic states increase from 10 to 40, the classification accuracy exhibits a gradual improvement, starting at 87% and nearly reaching 90% after just 50 epochs with 40 states. Furthermore,**

it is noteworthy that employing a greater number of synaptic states results in shorter training times. In Fig. 6f, we represented the confusion matrix that demonstrates the classification accuracy for each label.”

Comment: Many device physicists who would be interested in the paper will not necessarily be familiar with machine learning and as such the authors should try to make the terminology as accessible as possible. The authors should be careful for instance that terms / concepts (e.g. epoch, synaptic weight, confusion matrix) are actually explained when they are first introduced, so that readers can easily understand the impact of the demonstration rather than having to look up terms themselves.

Response: We extend our appreciation to the reviewer for their feedback aimed at improving the clarity of this paper. We have incorporated all the suggested changes and hope that the reviewer will be pleased with our modifications. See new Figure 6e and caption of Figure 6e in pages 16 and 17.

Comment: On a more minor note, in 6f it was not immediately clear to me that the values here being referred to are the handwritten digits – I would change the axis labels to something like predicted / actual handwritten digit.

Response: We have made changes to the labels in Figure 6f as per your recommendation. See new Figure 6f in page 16.

Reviewer #3 (Remarks to the Author):

Comment: The authors have conducted supplementary experiments and simulations to support their conclusions regarding the device's working mechanism and its neuromorphic applications. However, the two-layer ANN simulation, relying on P-D curves for MNIST dataset recognition, lacks novelty as it has already been extensively reported. Moreover, the operational mechanism of the non-volatile heterostructure spectrometer remains unclear, and certain explanations in the rebuttal letter is inconsistent with those presented in the initial manuscript. Therefore, I cannot recommend the publication of this paper in Nature Communications unless the authors elucidate the device's working mechanism and resolve the raised concerns outlined below.

In the rebuttal letter, the authors claim that the presence of Re vacancies induces trap states in ReSe₂ flake, which leads to the photogating and non-volatile memory effect. However, this clarification is inconsistent with the content in the manuscript, where it is stated that both the trap states in ReSe₂ and SnS₂ flakes could contribute to the observed effect. According to the manuscript, the photo-generated holes and electrons are trapped in SnS₂ and ReSe₂

respectively, leading to the photogating effect. To validate the defects induced photogating effect in the heterostructure, it is crucial to conduct the first-principle calculation, which could elucidate the distribution of trap states both in SnS₂ and ReSe₂ and their impact on photo-carrier transport and storage in the heterostructure. Moreover, the characterizations including TEM and XPS should be conducted to confirm the types and concentration of defects in SnS₂ and ReSe₂, which are more precise and reliable than the EDX in the rebuttal letter. Additionally, apart from the defects present in the individual flakes, the interfacial trap states are generally observed in 2D heterostructures during stacking process, which might also trap photo-generated carriers and prolong the carrier lifetime. The authors should also clarify whether such interfacial traps contribute to the photogating effect in their devices.

Response: Thank you very much for raising these good points. According to your suggestion, we performed TEM analysis and XPS characterization, and also conducted the first-principle calculations.

Figure R2. (a) High-resolution TEM image of mechanically exfoliated SnS₂ and (b) ReSe₂ layers, where red arrows highlight the positions of vacancies. The insets show the corresponding SAED images. The scale bars are 2 nm. High-resolution XPS spectra (c) Sn 3d_{3/2} and Sn 3d_{5/2}, (d) S 2p_{1/2} and S 2p_{3/2} region of SnS₂. (e) High-resolution XPS spectra Re 3f_{7/2} and Re 3f_{5/2}, (f) Se 3d_{5/2} and Se 3d_{3/2} region of ReSe₂.

Figures R2a and R2b show high-resolution TEM images of mechanically exfoliated SnS₂ and ReSe₂ layers. One can see that both exfoliated nanosheets are highly crystalline in nature. However, some vacancies could be observed (red arrows). To determine the type and estimate the concentration of vacancies, X-ray photoelectron spectroscopy (XPS) was carried out on SnS₂ and ReSe₂ to ascertain both their different oxidation states and stoichiometry, crucial factors in determining their fundamental properties, which can influence the device working principle. Figures R2c and R2d show the core level spectra for Sn3d and S2p, respectively. The binding energy values were calibrated with reference to the carbon 1s peak, which was set at 284.7 eV. The Sn 3d_{5/2} transition was deconvoluted into two values at binding energy (BE) values of 485.3 eV and 486.4 eV, corresponding to Sn oxidation states of +2 and +4. The sulfur 2p core-level XPS spectrum showed two peaks at BEs of 162.7 eV and 163.7 eV. Finally, the atomic weight percentages obtained from the deconvoluted states reveal a stoichiometry of SnS_{1.98}, indicating ~2% sulfur vacancies in our SnS₂. Figures R2e and R2f depict the core-level spectrum of Re4f and Se 3d. In the case of the Re 4f core-level spectrum, it typically exhibits two peaks corresponding to Re 4f_{7/2} and Re 4f_{5/2}, with binding energy values of 42.1 eV and 44.5 eV, respectively. For the core-level XPS spectra of Se 3d, to obtain a satisfactory fitting, four peaks were used instead of two corresponding to Se3d_{5/2} and Se3d_{3/2}. Furthermore, the atomic weight percentages obtained from the deconvoluted states reveal a stoichiometry of ReSe_{2.04}, indicating ~2.4% Re vacancies. The spectrum complexity and the additional peak observed may be related to the production of electron-deficient Se^{(2-δ)-} sites in our Se-rich ReSe_{2+x} materials.

Figure R3. (a) Atomic structures of ideal ReSe₂, (b) ReSe₂ with Re vacancy, (c) ideal SnS₂, and (d) SnS₂ with S vacancy, respectively. (e) Electronic structures of ideal ReSe₂, (f) ReSe₂ with Re vacancy, (g) ideal SnS₂, and (h) SnS₂ with S vacancy, respectively.

To understand photogating effect induced by defects in the heterostructure, we conducted first-principle calculations, as per your suggestion. Figures R3a-d show the atomic structures for those calculations. We consider a 2x2x2 super cell structure for ideal ReSe₂ and a 3x3x2 super structure for ideal SnS₂. To investigate the effect of Re vacancies and S vacancies, one Re atom and one S atom were removed from the ideal ReSe₂ and SnS₂ structures respectively. Figure R3e-h plots the corresponding calculated electronic structures. Note, the top of the valance bands (VB) was shifted to zero. Interestingly, when one Re atom was removed, three defect bands appeared and were labeled as DF_{R1} (0.26 eV below bottom of conduction band (CB)), DF_{R2} (0.49 eV below the bottom of CB), and DF_{R3} (0.37 eV above the top of VB). For SnS₂ with a S vacancy, only one defect band appeared (DF_S) and is located 0.7 eV below the bottom of the CB.

Figure R4 The transfer characteristics of (a) SnS₂ and (b) ReSe₂ in darkness at $V_{ds} = 2$ V. (c) Schematic of the electronic structure of SnS₂ and (d) ReSe₂. (e) Calculated built-in potential of the total heterostructure (V_{bihet} , green line), ReSe₂ (V_{biR} , blue line) and SnS₂ (V_{bis} , red line). (f) Schematic of energy band of SnS₂/ReSe₂ heterostructures after contact.

To further understand the defect induced photogating effect in the heterostructure, we calculated the Fermi energies for SnS₂, ReSe₂, and a SnS₂/ReSe₂ heterostructure. Figures R4a and R4b show the transfer characteristics of SnS₂ and ReSe₂ respectively. The extrinsic field-effect mobility, μ_{FE} , of the electrons in SnS₂ and ReSe₂ can be calculated from the equation

$$\mu_{FE} = \frac{g_m L}{W C_{ox} V_{ds}} \text{ at a constant drain-source voltage } V_{ds} = 2 \text{ V, where } g_m = \left. \frac{\partial I_{ds}}{\partial V_g} \right|_{V_{ds}=const}$$

is the transconductance of the field effect transistor, L is channel length (20 μm for both SnS₂ and ReSe₂), W is the channel width (27 μm for SnS₂ and 36 μm for ReSe₂), $C_{ox} = 5.76 \times 10^{-5} \text{ F/m}^2$ is the capacitance of the gate oxide (600 nm thick SiO₂). Using the slope, *i.e.* g_m , obtained from the linear fit to the linear region of the transfer curve, we can get the $\mu_{FE}^{\text{SnS}_2} = 2.17 \text{ cm}^2 \cdot (\text{V} \cdot \text{s})^{-1}$, $\mu_{FE}^{\text{ReSe}_2} = 0.614 \text{ cm}^2 \cdot (\text{V} \cdot \text{s})^{-1}$. Using the calculated mobilities, we can obtain the Fermi Energies for SnS₂ (-4.38 eV) and ReSe₂ (-4.03 eV).

The Fermi energy of the SnS₂/ReSe₂ heterostructure was determined by requiring that the increased charges in the SnS₂ equaled the decrease in ReSe₂, for the initial Fermi level of ReSe₂ being higher than that in SnS₂, *i.e.*,

$$\Delta n_e = n_{e0S}(E_{F0S}) - n_{eS}(E_F) = -[n_{e0R}(E_{F0R}) - n_{eR}(E_F)], \quad (5)$$

where $n_{e0S(R)}$ is the initial electron density of SnS₂ (ReSe₂) with the initial Fermi energy $E_{F0S(R)}$, and $n_{eS(R)}$ is the electron density after charge transfer from ReSe₂ into SnS₂ with the Fermi energy E_F . The Fermi energy E_F and the transfer charge can be estimated by the following expressions

$$E_F = E_{F0R} + k_B T \times \ln \left(1 + \frac{\Delta n_e}{n_{e0R}} \right) \text{ and} \quad (6)$$

$$\Delta n_e = n_{e0S}(E_{F0S}) - N_{CS} e^{-\frac{E_{CS} - E_F}{k_B T}}, \quad (7)$$

N_{CS} is the effective conduction-band density of states, it takes the value $4.615 \times 10^{26} m^{-3}$, and E_{CS} ($-4.22 eV$) is the conduction band edge of SnS₂. Solving Equations 5-7 yields a Fermi energy of $-4.35 eV$. The built-in potentials V_{biS} and V_{biR} at SnS₂ and ReSe₂ are calculated by $V_{biS} = E_F - E_{F0S}$ and $V_{biR} = E_{F0R} - E_F$. The calculated results are shown in Figure R4e. The largest built-in potential is distributed on the ReSe₂ side, however, the SnS₂ side changes the most with increasing V_g . Moreover, the energy band of ReSe₂ bends upward and that of SnS₂ bends downward after contact. Figure R4f shows a schematic of the energy bands of the SnS₂/ReSe₂ heterostructures after contact. One can find that for SnS₂, DF_S is located below the Fermi energy and far above the VB of SnS₂. Thus, traps in the SnS₂ can be ignored. However, for ReSe₂, the energy distance between the bottom of DF_{R3} and the top of the VB is only $\sim 0.07 eV$. Moreover, due to the upward bending of the energy band, hole trapping plays a unique role in the overlapping region as excited holes will move to the overlapped region and electrons will move to the nonoverlapped region. Thus, under illumination, holes are excited and move to the overlapping region and some of the holes will be trapped by DF_{R3} , resulting a photogating effect, which would have the effect of enhancing the photocurrent. When the light is switched off, the trapped holes remain and sustain the photocurrent, resulting in a memory effect. We would like to stress that the number of holes trapped depends on V_{biR} . Increasing V_g , V_{biR} decreases, resulting in a decrease in the number of trapped holes, which is consistent with the experimental results. For photodetectors dominated by a photo-gating effect, the photocurrent (I_{ph}) can be written as $I_{ph} = \frac{\partial I_{ds}}{\partial V_g} \Delta V_g$, where ΔV_g is the local gate voltage generated by photoexcited carrier trapping at the interface. In Figure 1m of main text, the calculated ΔV_g is shown to decrease with increasing gate voltage.

Fig. R5. Schematic diagram of the band profile (a) before and (b) after illumination.

About the neutral traps (NT) at the interface, the upwardly bent band raises the energy position of NTs in the overlapping region above the Fermi level (Figure R5a). Under illumination, electrons of the NTs in the overlapping region are excited to the conduction band and transported to the lower energy side (non-overlapping SnS₂ region) as shown in Figure R5b. In this case, the NTs become positively charged, lowering the transport barrier at the interface and thus increasing the photocurrent. When the light is switched off, the partially charged NTs remain at a position above the Fermi level, thereby sustaining the photocurrent, again producing a memory effect. Thus, neutral traps at the interface also result in photogating and memory effects.

This important piece of information has been added to the main text in revision as well. See from line 6 to line 24 in page 6 in main text and also Supplementary Note 1 in supporting information. **“TEM analysis and XPS characterization suggest that our SnS₂ possesses ~2% sulfur vacancies, while the ReSe₂ has ~2.4 % Re vacancies (Figs. 1c and 1d and Supplementary Fig. 6). First-principle calculations reveal Re vacancies result in the appearance of three defect bands labeled DF_{R1} (0.26 eV below bottom of conduction band (CB)), DR_{R2} (0.49 eV below the bottom of CB), and DF_{R3} (0.37 eV above the top of the valence band (VB)). In the case of SnS₂ with sulfur vacancies, a single defect band (DF_s) emerges, positioned 0.7 eV below the bottom of the CB (Supplementary Fig. 7). Moreover, due to the upward bending of the energy band, hole trapping plays a unique role in the overlapping region as excited holes will move to the overlapped region and electrons move to the nonoverlapped region (Fig. 1l and Supplementary Fig. 8). Thus, under illumination, holes are excited and move to the overlapping region and some of the holes will be trapped by DF_{R3}, resulting a photogating effect, which would have the effect of enhancing the photocurrent. When the light is switched off, the trapped holes remain and sustain the photocurrent, resulting in a memory effect. This is also key to the electrically tunable memory effect, discussed further in the paper. Moreover, the interfacial trap states, such as neutral traps (NT), are generally observed in 2D heterostructures during the stacking process, which also result in photogating and memory effects (Supplementary Fig. 9). Details of the calculation can be found in Supplementary Note 1. ”**

Comment: To elucidate the peak current in the SnS₂/ReSe₂ spectrometer, the authors present a band alignment diagram involving three photovoltaic voltages generated in the ReS₂, overlapped SnS₂/ReSe₂, and SnS₂ regions under light illumination. However, it is still not clear how these photovoltaic voltages compete with each other as the wavelength changes, leading to the emergence of peak current. Also, why is the peak position only determined by the bandgap of SnS₂, as claimed by the authors in the rebuttal letter? To address these

uncertainties, the authors should provide a more quantitative analysis based on the experimental results and the proposed model.

Response: Thank you for highlighting this weakness and we agree with you upon reflection that the origin of the peak of photocurrent in spectrometer lacked clarity in the earlier version.

Fig. R6. (a) Schematic energy band and photocurrent of SnS₂/ReSe₂ heterostructures after contact. (b) Photocurrent of different areas in the heterostructure extracted from Figure 1h as a function of laser wavelength. (c) Calculated ΔV_g of the device as a function of laser wavelengths at different gate voltages.

The total photocurrent (I_{Ph}) has three main contributions: the photocurrents generated in the ReS₂ (I_R) region, SnS₂ regions (I_S), and the photocurrent generated in the overlapping region ($I_{interface}$) due to the photogating effect. As the Au electrodes have a much greater work function compared with those of SnS₂ and ReSe₂, ϕ_{B1} is greater than V_{biR} . Thus, I_S and I_R have opposite directions. Therefore, $I_{Ph} = I_S - I_R + I_{interface}$. Moreover, $I_{interface} = \frac{\partial I_{ds}}{\partial V_g} \Delta V_g$

is almost constant when far from the photocurrent peak (15 V in Figure R6c). Thus, the peak of photocurrent can be explained by $|I_S - I_R|$. Figure R6b plots the R-S and it does show a peak around 600 nm, which consistent with the photocurrent of the device. We can further write the photocurrents generated in the ReS₂ and SnS₂ regions as: $I_i = qG_i(\lambda)\tau_i\mu_iSV_{bii}/W_{Di}$, where I_i , $G_i(\lambda)$, τ_i , μ_i , V_{bii} , and W_{Di} are the photocurrent, photogeneration rate, lifetime, mobility, effective built-in potential, and effective depletion width in region i respectively. More, the built-in potential of the SnS₂ side decreases the most with increasing Vg and that in the ReSe₂ side remains quite stable (Figure R4e). In other words, the photocurrent decreases much fast with increasing Vg. Thus, the peak position of the photocurrent moves to longer wavelengths with increasing Vg, which is consistent with the experimental observations (Figure R6c). It looks like

the peak position only determined by the bandgap of SnS₂. To avoid possibly misleading readers, we have deleted this claim in the revision.

We also tried to provide a quantitative simulation to get the exact peak position of the photocurrent. But, since defect band evolve and the device is under light illumination, there is no suitable model to quantitatively describe it. However, R-S does show a peak around 600 nm and the decrease in built-in potential of the SnS₂ side with increasing V_g can explain the movement of the peak position of photocurrent to longer wavelengths with V_g.

This important piece of information has been added to the main text in revision as well. See from line 24 in page 6 to line 14 in page 7 in main text and also Supplementary Note 2 in supporting information. **“For photodetectors dominated by a photogating effect. the**

photocurrent in the interface ($I_{interface}$) can be written as $I_{interface} = \frac{\partial I_{ds}}{\partial V_g} \Delta V_g$, where ΔV_g

is the local gate voltage generated by photoexcited carrier trapping at the interface. In Fig. 1m, the calculated ΔV_g is shown to decrease with increasing gate voltage. We would like to stress that the number of holes trapped depends on built-in potential at ReSe₂ (V_{biR}). Increasing V_g, V_{biR} decreases, resulting in a decrease in the number of trapped holes, which is consistent with the experimental results. The total photocurrent (I_{Ph}) has three main contributions: the photocurrents generated in the ReSe₂ (I_R) and SnS₂ regions (I_S), and the photocurrent generated in the overlapping region ($I_{interface}$) due to the photogating effect. As the Au electrode has a much greater work function in comparison with both SnS₂ and ReSe₂, the built-in potential between Au and ReSe₂ (ϕ_{B1}) is greater than V_{biR} . Thus, I_S and I_R are in opposite directions. Therefore, $I_{Ph} = I_S -$

$I_R + I_{interface}$. Moreover, $I_{interface} = \frac{\partial I_{ds}}{\partial V_g} \Delta V_g$ is almost constant when far from the

photocurrent peak (15 V in Fig. 1m). Thus, the peak of photocurrent can be explained by $|I_S - I_R|$. Fig. 1j plots the R-S and it does show a peak around 600 nm, which is consistent with the photocurrent of the device. Moreover, the built-in potential of the SnS₂ side decreases the most with increasing V_g and that in the ReSe₂ side remains quite stable (Supplementary Fig. 8e). In other words, the photocurrent decreases much faster with increasing V_g (Supplementary Note 2). Thus, the peak position of the photocurrent moves to longer wavelength with increasing V_g, which is consistent with the experimental observation (Fig. 1m). ”

Comment: The inclusion of additional simulations based on a two-layer ANN lacks novelty in demonstrating the spectrometer’s potential in non-von Neumann architecture. I think that further investigation into the mechanism of the spectrometer is necessary to enhance the

insights gained from this work.

Response: In the revision, we clarified the defect induced photogating effect and provided a better understanding of the origin of the peak of photocurrent. We hope that the corrections are to your satisfaction and that you will consider the manuscript for publication in Nature Communications.

REVIEWERS' COMMENTS

Reviewer #2 (Remarks to the Author):

I commend the authors on their significant efforts, both experimental and in writing, to improve the explanations around the working principles of the device, and their clarification of the operation of the device in relation to the ANN that they present. Particularly with regard to the former, the work is now far more thorough, and in both cases there is greater clarity.

I still have minor concerns around the neural network demonstration, given that this is not in itself novel and as far as I can tell, there is no overlap of this functionality with that of the device working as a spectrometer (if this is not the case I would invite the authors to clarify in further edits of the manuscript).

However, I believe my previous comments have been sufficiently addressed for me to recommend publication at the editor's discretion.

Reviewer #3 (Remarks to the Author):

The authors have performed additional experiments and theoretical calculations to validate their claim regarding to the defects induced photogating and non-volatile memory effect. The working mechanism of the heterostructure spectrometer has also been further clarified. I think that the manuscript in its current version could be accepted for publication in Nature Communications.

Response to reviewers' comments

Reviewer #2 (Remarks to the Author):

Comment: I commend the authors on their significant efforts, both experimental and in writing, to improve the explanations around the working principles of the device, and their clarification of the operation of the device in relation to the ANN that they present. Particularly with regard to the former, the work is now far more thorough, and in both cases there is greater clarity.

I still have minor concerns around the neural network demonstration, given that this is not in itself novel and as far as I can tell, there is no overlap of this functionality with that of the device working as a spectrometer (if this is not the case I would invite the authors to clarify in further edits of the manuscript).

However, I believe my previous comments have been sufficiently addressed for me to recommend publication at the editor's discretion.

Response: We sincerely appreciate your positive feedback on our work and value the insightful opinions you have shared. We acknowledge the concern you raised regarding the direct demonstration of functional overlap in this work. It is important to note that we have previously addressed the issue of functional overlap in the main text on page 13. To quote directly, 'Moreover, the shape of long-term potentiation demonstrates a dependence on incident light, implying the possibility of reconstructing the spectrum of incident light with a defined number of memory cells (Fig. 7c).' In simpler terms, the combination of compressed sensing allows us to determine the minimum number of gate voltage values and, consequently, the minimum

number of device cells (pixels) required to reconstruct the full spectrum. Additionally, this allows the same memory cell to be used as a synapse for neuromorphic computing. We will diligently work on incorporating your valuable suggestion into our future research to provide a more comprehensive demonstration.

Reviewer #3 (Remarks to the Author):

Comment: The authors have performed additional experiments and theoretical calculations to validate their claim regarding to the defects induced photogating and non-volatile memory effect. The working mechanism of the heterostructure spectrometer has also been further clarified. I think that the manuscript in its current version could be accepted for publication in Nature Communications.

Response: We greatly appreciate your affirmation of this work and for raising many useful and valuable suggestions.